# Node Feature Forecasting in Temporal Graphs: an Interpretable Online Algorithm

**Aniq Ur Rahman**  *aniq.rahman@eng.ox.ac.uk*
**Justin P. Coon**  *justin.coon@eng.ox.ac.uk*
*Department of Engineering Science, University of Oxford, U.K.*

**Reviewed on OpenReview:** *https://openreview.net/forum?id=Teu1Blr2YJ*

## Abstract

In this paper, we propose an online algorithm `mspace` for forecasting node features in temporal graphs, which captures spatial cross-correlation among different nodes as well as the temporal auto-correlation within a node. The algorithm can be used for both probabilistic and deterministic multi-step forecasting, making it applicable for estimation and generation tasks. Evaluations against various baselines, including temporal graph neural network (TGNN) models and classical Kalman filters, demonstrate that `mspace` performs comparably to the state-of-the-art and even surpasses them on some datasets. Importantly, `mspace` demonstrates consistent performance across datasets with varying training sizes, a notable advantage over TGNN models that require abundant training samples to effectively learn the spatiotemporal trends in the data. Therefore, employing `mspace` is advantageous in scenarios where the training sample availability is limited. Additionally, we establish theoretical bounds on multi-step forecasting error of `mspace` and show that it scales linearly with the number of forecast steps $q$ as $\mathcal{O}(q)$. For an asymptotically large number of nodes $n$, and timesteps $T$, the computational complexity of `mspace` grows linearly with both $n$, and $T$, i.e., $\mathcal{O}(nT)$, while its space complexity remains constant $\mathcal{O}(1)$. We compare the performance of various `mspace` variants against ten recent TGNN baselines and two classical baselines, `ARIMA` and the `Kalman` filter across ten real-world datasets. Lastly, we have investigated the interpretability of different `mspace` variants by analyzing model parameters alongside dataset characteristics to jointly derive model-centric and data-centric insights. [Link to Code]

## 1 Introduction

Temporal graphs are a powerful tool for modelling real-world data that evolves over time. They are increasingly being used in diverse fields, such as recommendation systems (Gao et al., 2022), social networks (Deng et al., 2019), and transportation systems (Yu et al., 2018), to name a few. Temporal graph learning (TGL) can be viewed as the task of learning on a sequence of graphs that form a time series. The changes in the graph can be of several types: changes to the number of nodes, the features of existing nodes, the configuration of edges, or the features of existing edges. Moreover, a temporal graph can result from a single or a combination of these changes. The TGL methods can be applied to various tasks, such as regression, classification, and clustering, at three levels: node, edge, and graph (Longa et al., 2023).

In this work, we focus on node feature forecasting, also known as node regression, where the previous temporal states of a graph are used to predict its future node features. Forecasting is a fundamental problem in various domains, such as weather, finance, and traffic, enabling informed decision-making (Petropoulos et al., 2022), and the problem still remains relevant today in light of the advances in machine learning. In the context of temporal graphs, node feature forecasting exploits the structure of the evolving network assuming that the future value of a node is influenced by its neighbours (Huang et al., 2023).

In most temporal graph neural network (TGNN) models, the previous states are encoded into a super-state or dynamic graph embedding (Barros et al., 2021), guided by the graph structure. This dynamic embedding

is then used to forecast the future node features. Although TGNN models perform well, their interpretability is often overlooked, and their performance is not explained through the data. Furthermore, the relationship between the node features and the node or graph embeddings is not human-understandable. Furthermore, most embedding aggregation mechanisms impose a strong assumption that the neighbours influence a node in proportion to their edge weight (Wang et al., 2021).

TGNN methods (Li et al., 2018; Micheli & Tortorella, 2022; Wu et al., 2019; Fang et al., 2021; Liu et al., 2023) typically involve a training phase where the model learns from training data and is then deployed on test data without further training due to computational costs. If the test data distribution differs from the training data, an offline model cannot adapt (Wang et al., 2024). Therefore, when dealing with time-series data, it is crucial to use a lightweight online algorithm that can adapt to changes in data distribution while also performing forecasts. Moreover, TGNN models are typically trained to forecast a predetermined number of future steps. If we want to increase the number of forecast steps, even by one, the model needs to be reinitialized and retrained. Alternatively, the output can be fed back as input to the TGNN, extending the forecasting scale of the same model without additional training.

Inspired by the simplicity of Markov models, we define the state of a graph at a given time in an interpretable manner and propose a lightweight model that can be deployed without any training. The algorithm is designed with a mechanism to prioritize recent trends in the data over historical ones, allowing it to adapt to changes in data distribution.

**Contributions**   The contributions of our work are summarized as follows:

- We have proposed an online learning algorithm `mspace` for node feature forecasting in temporal graphs, which can sequentially predict the node features for $q \in \mathbb{N}$ future timesteps after observing only two past node features.

- The algorithm `mspace` can produce both probabilistic and deterministic forecasts, making it suitable for generative and predictive tasks.

- The root mean square error (RMSE) of $q$-step iterative forecast scales linearly in the number of steps $q$, i.e. $\text{RMSE}(q) = \mathcal{O}(q)$.

- For asymptotically large number of nodes $n$, and timesteps $T$, the computational complexity of `msapce` grows linearly with both $n$, and $T$, i.e., $\mathcal{O}(nT)$, while the space complexity is constant $\mathcal{O}(1)$.

- We have compared the performance of different variants of `mspace` against ten recent TGNN baselines, and two classical baselines `ARIMA`, and `Kalman` filter.

- We have evaluated `mspace` on four datasets for single-step forecasting and six datasets for multi-step forecasting.

- In addition to the evaluation on ten real-world datasets, we have proposed a technique to generate synthetic datasets that can aid in a more thorough evaluation of node feature forecasting methods. The synthetic datasets have the potential to serve as benchmark for future research.

- We have investigated the interpretability of different `mspace` variants by analyzing the model parameters along with the dataset characteristics to jointly derive model-centric and data-centric insights.

- To facilitate the reproducibility of results, the **code** is made available here.

**Notation**   We denote vectors with lowercase boldface $\boldsymbol{x}$, and matrices and tensors with uppercase boldface $\boldsymbol{X}$. Sets are written in calligraphic font such as $\mathcal{V}, \mathcal{U}, \mathcal{S}, \mathcal{C}$, with the exception of graphs $\mathcal{G}$, and queues $\mathcal{Q}$. The operator $\succ$ is used in two contexts: $\boldsymbol{x} \succ \boldsymbol{0}$ is an element wise positivity check on the vector $\boldsymbol{x}$, and $\mathbf{A} \succ \boldsymbol{0}$ indicates that the matrix $\mathbf{A}$ is positive definite. $\mathbb{I}(\cdot)$ is the indicator function, and $[m] \triangleq \{1, 2, \cdots, m\}$ for any $m \in \mathbb{N}$. We denote the distributions of continuous variables by $p(\cdot)$, and of discrete variables by $P(\cdot)$. The

statement $\boldsymbol{x} \sim p$ means that $\boldsymbol{x}$ is sampled from $p$. The Hadamard product operator is denoted by $\odot$ while the Kronecker product operator is denoted by $\otimes$. The trace of a matrix $\mathbf{A}$ is written as $\operatorname{tr}(\mathbf{A})$.

We denote the neighbours of a node $v$ for an arbitrary number of hops as $\mathcal{U}_v$. The neighbours of node $v$ up to $K$ number of hops is defined as follows. Let $\mathbf{N} = \sum_{k \in [K]} \mathbf{A}^k$, then $\mathcal{U}_v = \{u : \mathbf{N}_{v,u} > 0, \forall u \in \mathcal{V}\}$. Since $\mathbf{A}_{v,v} = 1$, $v \in \mathcal{U}_v$. We introduce the operator $\langle \cdot \rangle$ to arrange the nodes in a set $\mathcal{U}$ in ascending numerical order of the node IDs. When another set or vector is super-scripted with $\langle \mathcal{U} \rangle$, the elements within that set or vector are filtered and arranged as per $\langle \mathcal{U} \rangle$.

A Markov chain is represented using $\mathfrak{Z}$ with different subscripts for identification. The transition kernel of a Markov chain is denoted as $\mathbf{P}$ with $\mathbf{P}_{\boldsymbol{a},\boldsymbol{b}}$ representing the probability of transitioning from state $\boldsymbol{a}$ to $\boldsymbol{b}$.

**Organization** In Sec. 2 we formulate the problem of node feature forecasting and also a propose a model to solve it. In Sec. 3 we expand upon the solution and present it as an algorithm. We discuss the related works in Sec. 4 and present the results on single-step and multi-step node feature forecasting in Sec. 5. In Sec. 6 we discuss the interpretability of the proposed algorithm and then discuss the limitations in Sec. 7. Finally, we conclude in Sec. 8.

## 2 Methodology

**Problem Formulation** A discrete-time temporal graph is defined as $\{\mathcal{G}_t = (\mathcal{V}, \mathcal{E}, \boldsymbol{X}_t) : t \in [T]\}$, where $\mathcal{V} = [n]$ is the set of nodes, $\mathcal{E} \subseteq \mathcal{V} \times \mathcal{V}$ is the set of edges, and $\boldsymbol{X}_t \in \mathbb{R}^{n \times d}$ is the node feature matrix at time $t$. The set of edges $\mathcal{E}$ can alternatively be represented by the adjacency matrix denoted as $\boldsymbol{A} \in \{0,1\}^{n \times n}$. The node feature vector is denoted by $\boldsymbol{x}_t(v) \in \mathbb{R}^d$ such that $\boldsymbol{X}_t = \left[\boldsymbol{x}_t(v)\right]_{v \in \mathcal{V}}^\top$, and we refer to the first-order differencing (Shumway & Stoffer, 2017) of a node feature vector as **shock**. For a node $v \in \mathcal{V}$ we define the shock at time $t$ as $\boldsymbol{\varepsilon}_t(v) \triangleq \boldsymbol{x}_t(v) - \boldsymbol{x}_{t-1}(v)$. The shock of the nodes in an ordered set $\mathcal{U}$ at time $t$ is denoted by $\boldsymbol{\varepsilon}_t^{\langle \mathcal{U} \rangle} \in \mathbb{R}^{|\mathcal{U}|d}$. The shock at time $t$ for an arbitrary set of nodes is $\boldsymbol{\varepsilon}_t$.

To address the problem, we make certain assumptions. The first is a Markov assumption, stated as follows:

**Assumption 2.1.** The shocks $\{\boldsymbol{\varepsilon}_1, \boldsymbol{\varepsilon}_2, \boldsymbol{\varepsilon}_3 \cdots \boldsymbol{\varepsilon}_T\}$ is assumed to be sampled from a continuous-state Markov chain defined on $\mathbb{R}^{md}$ for some $m \in [n]$, such that $p(\boldsymbol{\varepsilon}_{t+1} \mid \boldsymbol{\varepsilon}_t, \boldsymbol{\varepsilon}_{t-1}. \cdots) = p(\boldsymbol{\varepsilon}_{t+1} \mid \boldsymbol{\varepsilon}_t)$.

Although the Markov assumption can be extended to higher orders, in this work, we consider only a first-order Markov chain, which limits the model's ability to capture long-range dependencies in the data. In theory, a continuous-state Markov chain has infinite number of states which makes it impossible to learn the transition kernel from limited samples without additional assumptions. To circumvent this, *linear dynamical systems* and *autoregressive models* are used in the literature (Barber, 2012) where the next state is determined through a function of the current state.

Let $p(\boldsymbol{\varepsilon}' \mid \boldsymbol{\varepsilon})$ denote the transition probability $\boldsymbol{\varepsilon} \to \boldsymbol{\varepsilon}'$ in a continuous-state Markov chain $\mathfrak{Z}_0$ defined over a set $\mathcal{C}$. A discrete-state Markov chain $\mathfrak{Z}_1$ defined over finite set $\mathcal{S}$ with transition probability $\mathbf{P}_{\boldsymbol{s},\boldsymbol{s}'}$ can be constructed from $p(\boldsymbol{\varepsilon}' \mid \boldsymbol{\varepsilon})$ through a mapping[1] $\Psi : \mathcal{C} \to \mathcal{S}$ as

$$\mathbf{P}_{\boldsymbol{s},\boldsymbol{s}'} = \frac{\iint_{\mathcal{C}\,\mathcal{C}} p(\boldsymbol{\varepsilon}' \mid \boldsymbol{\varepsilon}) p(\boldsymbol{\varepsilon}) \, \mathbb{I}(\Psi(\boldsymbol{\varepsilon}) = \boldsymbol{s}) \, \mathbb{I}(\Psi(\boldsymbol{\varepsilon}') = \boldsymbol{s}') \, d\boldsymbol{\varepsilon} \, d\boldsymbol{\varepsilon}'}{\iint_{\mathcal{C}\,\mathcal{C}} p(\boldsymbol{\varepsilon}' \mid \boldsymbol{\varepsilon}) p(\boldsymbol{\varepsilon}) \, \mathbb{I}(\Psi(\boldsymbol{\varepsilon}) = \boldsymbol{s}) \, d\boldsymbol{\varepsilon} \, d\boldsymbol{\varepsilon}'}. \tag{1}$$

For a continuous-state Markov chain sample $\{\boldsymbol{\varepsilon}_1, \boldsymbol{\varepsilon}_2, \cdots \boldsymbol{\varepsilon}_T\}$, we can estimate $\mathbf{P}$ directly from $\{\Psi(\boldsymbol{\varepsilon}_1), \Psi(\boldsymbol{\varepsilon}_2), \cdots \Psi(\boldsymbol{\varepsilon}_T)\}$ without the need of $p(\boldsymbol{\varepsilon}' \mid \boldsymbol{\varepsilon})$. Now, consider a *random function* $\Omega : \mathcal{S} \to \mathcal{C}$, such that: (a) $\Psi(\boldsymbol{\varepsilon}) = \boldsymbol{s}$, (b) $\Psi(\boldsymbol{\varepsilon}') = \boldsymbol{s}'$, (c) $\boldsymbol{\varepsilon}' = \Omega(\boldsymbol{s})$, from which follows $p(\Omega(\boldsymbol{s})) = p(\boldsymbol{\varepsilon}' \mid \boldsymbol{s})$.

The approximate transition kernel $\hat{\mathbf{P}}$ due to $(\Psi, \Omega)$ can be written as:

$$\hat{\mathbf{P}}_{\boldsymbol{s},\boldsymbol{s}'} = \int_{\{\boldsymbol{\varepsilon}' \in \mathcal{C} : \Psi(\boldsymbol{\varepsilon}') = \boldsymbol{s}'\}} p(\boldsymbol{\varepsilon}' \mid \boldsymbol{s}) \, d\boldsymbol{\varepsilon}' = \int_{\mathcal{C}} p(\Omega(\boldsymbol{s})) \, \mathbb{I}(\Psi(\boldsymbol{\varepsilon}') = \boldsymbol{s}') \, d\boldsymbol{\varepsilon}'. \tag{2}$$

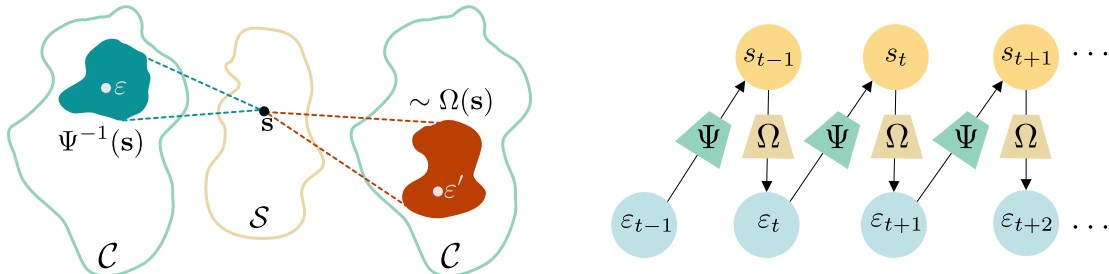

Figure 1: (left) state and sampling functions visualized, (right) Markov approximation.

In Fig. 1 (left) we depict the functions $\Psi$ mapping from continuous space in $\mathcal{C}$ to a discrete space $\mathcal{S}$. We also depict $\Omega$ mapping from $\mathcal{S}$ to $\mathcal{C}$. In a red patch we show the range of $\Omega(s)$, and in the green patch we show the domain of $\Psi(s)$. In Fig. 1 (right), we visualize Assumption 2.1 wherein the shocks evolve as a Markov chain through the functions $\Psi, \Omega$.

We refer to $\Psi$ as the **state function**, and $\Omega$ as the **sampling function**. The approximated Markov chain defined over $\mathcal{S}$ resulting from $(\Psi, \Omega)$ is denoted as $\hat{\mathfrak{Z}}(\Psi, \Omega)$, with $p(\hat{\varepsilon}' \mid \varepsilon) = p(\Omega \circ \Psi(\varepsilon))$. Ideally, the goal is to find the pair of functions $(\Psi, \Omega)$ such that: (a) $\mathbf{P}_{s,s'} = \hat{\mathbf{P}}_{s,s'} \, \forall s, s' \in \mathcal{S}$, (b) $p(\varepsilon'|\varepsilon) = p(\Omega \circ \Psi(\varepsilon)) \, \forall \varepsilon \in \mathcal{C}$. However, in practice this is quite ambitious as the state and sampling functions will induce some error in the encoding and decoding process. Therefore, we frame the problem as follows.

The sequence of shocks drawn from the original Markov chain $\mathfrak{Z}_0$ is represented as $\{\varepsilon_t : t \in [T]\} \sim \mathfrak{Z}_0$. Then, for each $\varepsilon_t$ we generate a sequence of $q$ future shocks using the Markov chain $\hat{\mathfrak{Z}}(\Psi, \Omega)$ as

$$\hat{\varepsilon}_{t+j} = (\Omega \circ \Psi)^j(\varepsilon_t), \quad \forall t \in [T-q], j \in [q].$$

The problem is to design $\Psi, \Omega$ such that $\left\| \sum_{j \in [k]} \varepsilon_{t+j} - \hat{\varepsilon}_{t+j} \right\|^2$ is minimized $\forall k \in [q], t \in [T-q]$, which can be written alternatively as:

**Problem 2.1** ($q$-step node feature forecasting). Design the state and sampling functions $\Psi, \Omega$ such that

$$\min \sum_{t \in [T-q]} \sum_{k \in [q]} \left\| \sum_{j \in [k]} \varepsilon_{t+j} - (\Omega \circ \Psi)^j \varepsilon_t \right\|^2. \tag{3}$$

In a deep learning context, both $\Psi$ and $\Omega$ would typically be neural networks trained directly using the objective in Problem 2.1. In this work, however, we explicitly define $\Psi$ and $\Omega$ and learn their parameters through the same objective.

**Proposed Model** Instead of creating a single model to approximate $p\left(\varepsilon_{t+1}^{\langle \mathcal{V} \rangle} \mid \varepsilon_t^{\langle \mathcal{V} \rangle}\right)$, we create a model for each node $v \in \mathcal{V}$ to approximate $p\left(\varepsilon_{t+1}^{\langle \mathcal{U}_v \rangle} \mid \varepsilon_t^{\langle \mathcal{U}_v \rangle}\right)$ where $\mathcal{U}_v$ denotes the neighbours of node $v$ within a certain number of hops. We present this in the following assumption.

**Assumption 2.2.** The shock of node $v$ at time $t + 1$ can be estimated from the shock of its neighbouring nodes in $\mathcal{U}_v$ at time $t$.

While $\varepsilon_t(u')$ for any node $u' \notin \mathcal{U}_v$ may help in estimating $\varepsilon_{t+1}(v)$, we assume that enough information is already conveyed by the nodes in $\mathcal{U}_v$ that the impact of considering node $u'$ would be minimal. It must be noted that $\mathcal{U}_v$ denotes the neighbours of node $v$ up to an arbitrary number of hops, therefore if we consider $\mathcal{U}_v$ to mean $k$ hops, then all the nodes that neighbours $v$ with $1, 2, \cdots, k$ hops are all in $\mathcal{U}_v$ and their impact

---

[1]For example, $\mathcal{C}$ is the set $\mathbb{R}^m$ for some integer $m$, and $\mathcal{S}$ is the set $\{1, -1\}^m$ or simply $\{1, 2, \cdots, L\}$ for some integer $L$.

is considered. Assumption 2.2 is important to create a scalable model, because in a connected graph every node will be correlated with every other node which will make the state space prohibitively large.

We propose two variants of the **state function**, one which captures the characteristics of the shock $\Psi_{\mathtt{S}}$, and the other which is concerned with the timestamps $\Psi_{\mathtt{T}}$ and captures seasonality.

- $\Psi_{\mathtt{S}} : \mathbb{R}^{|\mathcal{U}|d} \to \{-1, 1\}^{|\mathcal{U}|d}, \quad \Psi_{\mathtt{S}}(\boldsymbol{\varepsilon}^{\langle\mathcal{U}\rangle}) = \mathrm{sign}(\boldsymbol{\varepsilon}^{\langle\mathcal{U}\rangle}).$

- $\Psi_{\mathtt{T}} : \mathbb{N} \to \{0, 1, \cdots \tau_0 - 1\}, \quad \Psi_{\mathtt{T}}(t) = t \bmod \tau_0$, where $\tau_0 \in \mathbb{N}$ is the time period.

We also define two variants of the **sampling function**:

- deterministic $\Omega_\mu(\boldsymbol{s}) = \boldsymbol{\mu}(\boldsymbol{s}), \forall \boldsymbol{s} \in \mathcal{S}.$

- probabilistic $\Omega_{\mathcal{N}}(\boldsymbol{s}) \sim \mathcal{N}(\boldsymbol{\varepsilon}'; \boldsymbol{\mu}(\boldsymbol{s}), \boldsymbol{\Sigma}(\boldsymbol{s})), \forall \boldsymbol{s} \in \mathcal{S}.$

In $\Psi_{\mathtt{S}}$, we binarize the shock values of a node and its neighbors, creating a vector that indicates whether each value has increased or decreased. This vector serves as the state on which the next shock value is conditioned. Similarly, in $\Psi_{\mathtt{T}}$, we transform the timestamp into an integer based on a predefined time period, then use this integer to condition the next shock value. A comprehensive explanation of the state functions is provided in Sec. 6. The proposed model is presented as an online algorithm and discussed in detail in the following section.

## 3   Algorithm

We name our algorithm `mspace` with a suffix specifying the state and sampling functions. For example, `mspace-S`$\mathcal{N}$ represents the algorithm with state function $\Psi_{\mathtt{S}}$, and sampling function $\Omega_{\mathcal{N}}$. For each node $v \in \mathcal{V}$, we approximate $p(\boldsymbol{\varepsilon}_{t+1}^{\langle\mathcal{U}_v\rangle} \mid \Psi_{\mathtt{S}}(\boldsymbol{\varepsilon}_t^{\langle\mathcal{U}_v\rangle}) = \boldsymbol{s})$ as a Gaussian distribution with mean vector $\boldsymbol{\mu}_v(\boldsymbol{s}) \in \mathbb{R}^{|\mathcal{U}_v|d}$ and covariance matrix $\boldsymbol{\Sigma}_v(\boldsymbol{s}) \in \mathbb{R}^{|\mathcal{U}_v|d \times |\mathcal{U}_v|d}$ indexed by the state $\boldsymbol{s} \in \{-1, 1\}^{|\mathcal{U}_v|d}$. The parameters $\boldsymbol{\mu}_v(\boldsymbol{s}), \boldsymbol{\Sigma}_v(\boldsymbol{s})$ are learnt through maximum likelihood estimation (`MLE`). For each node $v \in \mathcal{V}$, and state $\boldsymbol{s}$ we maintain a **queue** $\mathcal{Q}_v(\boldsymbol{s})$ of maximum size $M$ in which the shocks succeeding a given state $\boldsymbol{s}$ are collected. The `MLE` solution is calculated as $\boldsymbol{\mu}_v(\boldsymbol{s}) \leftarrow \mathrm{mean}(\mathcal{Q}_v(\boldsymbol{s}))$, and $\boldsymbol{\Sigma}_v(\boldsymbol{s}) \leftarrow \mathrm{covariance}(\mathcal{Q}_v(\boldsymbol{s}))$.

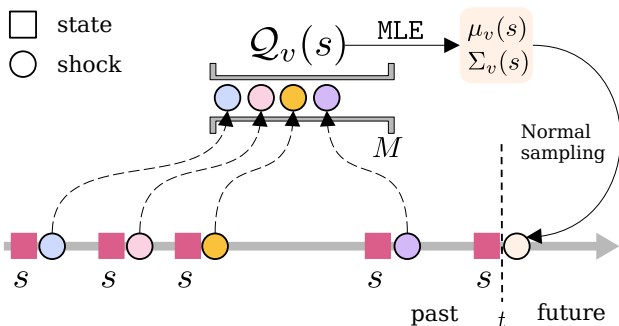

Figure 2: Operation of a queue.

The use of a fixed-size queue (see Fig. 2) ensures that the model prioritises recent data over historical data, thereby allowing the system to adapt to prevailing trends.

As `mspace` is an online algorithm, we might encounter unobserved states for which the queue is empty, and therefore cannot employ `MLE`. To facilitate *inductive inference*, as a state $\boldsymbol{s}_t$ is encountered, we find the state $\boldsymbol{s}^* \in \mathcal{S}_v$ which is the closest to $\boldsymbol{s}_t$, i.e., $\boldsymbol{s}^* \leftarrow \arg\min_{\boldsymbol{s} \in \mathcal{S}_v} \|\boldsymbol{s} - \boldsymbol{s}_t\|$, where $\mathcal{S}_v$ is the set of states observed before time $t$.

---

**Algorithm 1** mspace-S$\mathcal{N}$

---

**Input** $\mathcal{G} = (\mathcal{V}, \mathcal{E}, \boldsymbol{X})$, $r \in [0, 1)$, $q, M$
**Output** $\hat{\varepsilon}_t(v)$, $\forall v \in \mathcal{V}, t \in [\lfloor r \cdot T \rfloor, T]$
1: $\boldsymbol{\varepsilon}_t \leftarrow \boldsymbol{x}_t - \boldsymbol{x}_{t-1}, \quad \forall t \in [T]$

    *Offline training* (A)
2: **for** $t \in [\lfloor r \cdot T \rfloor]$ **do**
3:    **for** $v \in \mathcal{V}$ **do**
4:       $\boldsymbol{s}_t \leftarrow \Psi_{\text{S}}\left(\boldsymbol{\varepsilon}_t^{\langle \mathcal{U}_v \rangle}\right)$
5:       $\mathcal{S}_v \leftarrow \mathcal{S}_v \cup \{\boldsymbol{s}_t\}$
6:       $\mathcal{Q}_v(\boldsymbol{s}_t) \leftarrow$ enqueue $\boldsymbol{\varepsilon}_{t+1}^{\langle \mathcal{U}_v \rangle}$
7:    **end for**
8: **end for**

9: **for** $v \in \mathcal{V}$ **do**
10:   $\boldsymbol{\mu}_v(\boldsymbol{s}) \leftarrow \text{mean}(\mathcal{Q}_v(\boldsymbol{s})), \forall \boldsymbol{s} \in \mathcal{S}_v$
11:   $\boldsymbol{\Sigma}_v(\boldsymbol{s}) \leftarrow \text{covariance}(\mathcal{Q}_v(\boldsymbol{s})), \forall \boldsymbol{s} \in \mathcal{S}_v$
12: **end for**

    *Online learning* (B)
13: **for** $t \in [\lfloor r \cdot T \rfloor, T - q]$ **do**
14:   **for** $v \in \mathcal{V}$ **do**
15:     $\boldsymbol{s}_t \leftarrow \Psi_{\text{S}}\left(\boldsymbol{\varepsilon}_t^{\langle \mathcal{U}_v \rangle}\right)$
16:     $\boldsymbol{s}^* \leftarrow \arg\min_{\boldsymbol{s} \in \mathcal{S}_v} \|\boldsymbol{s} - \boldsymbol{s}_t\|$
17:     $\hat{\boldsymbol{\varepsilon}}_{t+1}^{\langle \mathcal{U}_v \rangle} \sim \mathcal{N}(\boldsymbol{\varepsilon}; \boldsymbol{\mu}_v(\boldsymbol{s}^*), \boldsymbol{\Sigma}_v(\boldsymbol{s}^*))$
18:     **for** $k \in [q] \setminus \{1\}$ **do**
19:       $\boldsymbol{s}^* \leftarrow \arg\min_{\boldsymbol{s} \in \mathcal{S}_v} \left\|\boldsymbol{s} - \Psi\left(\hat{\boldsymbol{\varepsilon}}_{t+k-1}^{\langle \mathcal{U}_v \rangle}\right)\right\|$
20:       $\hat{\boldsymbol{\varepsilon}}_{t+k}^{\langle \mathcal{U}_v \rangle} \sim \mathcal{N}(\boldsymbol{\varepsilon}; \boldsymbol{\mu}_v(\boldsymbol{s}^*), \boldsymbol{\Sigma}_v(\boldsymbol{s}^*))$
21:     **end for**
22:     $\hat{\boldsymbol{\varepsilon}}_{t+k}(v) \leftarrow \hat{\boldsymbol{\varepsilon}}_{t+k}^{\langle \mathcal{U}_v \rangle}(v), \quad \forall k \in [q]$
23:     Update $\mathcal{S}_v, \mathcal{Q}_v; \boldsymbol{\mu}_v(\boldsymbol{s}), \boldsymbol{\Sigma}_v(\boldsymbol{s}), \forall \boldsymbol{s} \in \mathcal{S}_v$
24:   **end for**
25: **end for**

---

**Example** For the purpose of explaining mspace-S$\mathcal{N}$ we consider an example with two nodes $n = 2$, and feature dimension $d = 1$. In Fig. 3 we first show the shock vector $\boldsymbol{\varepsilon}_t \in \mathbb{R}^2$. The state of shock $\boldsymbol{\varepsilon}_t$, denoted by $\Psi(\boldsymbol{\varepsilon}_t)$ is marked in $\mathcal{S} \in \{-1, 1\}^2$. Corresponding to this state, we have a Gaussian distribution $\mathcal{N}(\boldsymbol{\varepsilon}; \boldsymbol{\mu}(\Psi(\boldsymbol{\varepsilon}_t)), \boldsymbol{\Sigma}(\Psi(\boldsymbol{\varepsilon}_t)))$ depicted as an ellipse. The next shock $\boldsymbol{\varepsilon}_{t+1}$ is sampled from this distribution. This distribution is updated as we gather more information over time. The volume of the Gaussian density in a quadrant is equal to the probability of the next shock's state being in that quadrant, i.e., the transition kernel $\hat{\mathbf{P}}_{\boldsymbol{s}, \boldsymbol{s}'} = \int_{\boldsymbol{s}' \odot \boldsymbol{\varepsilon} \succ \boldsymbol{0}} \mathcal{N}(\boldsymbol{\varepsilon}; \boldsymbol{\mu}(\boldsymbol{s}), \boldsymbol{\Sigma}(\boldsymbol{s})) \, d\boldsymbol{\varepsilon}$. Therefore, mspace-S$\mathcal{N}$ can be viewed as a Markov chain whose transition function is a multivariate Gaussian.

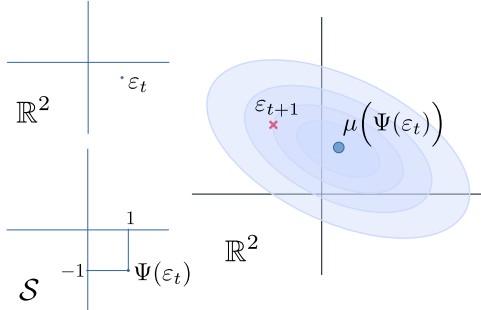

Figure 3: Shock Distribution.

## 4 Related Works

**Correlated Time Series Forecasting** A set of $n$ time series data denoted as $\boldsymbol{x}_t(v), \forall v \in [n], t \in [T]$ is assumed to exhibit spatio-temporal correlation (Wu et al., 2021a; Lai et al., 2023). The correlations can then be discerned from the observations to perform forecasting. The correlated time series (CTS) data can be viewed as a temporal graph $\mathcal{G} = (\boldsymbol{X}_t, \boldsymbol{A})$, with $\boldsymbol{X}_t \triangleq \left[\boldsymbol{x}_t(v)\right]_{v \in [n]}$ where the *spatial correlation* between $\boldsymbol{x}_t(u)$ and $\boldsymbol{x}_t(u)$ is quantified as the edge weight $\mathbf{A}_{u,v}$, and $\mathbf{A}_{u,u}$ signifies the *temporal correlation* within $\boldsymbol{x}_t(u)$. The architecture of existing CTS forecasting methods consist of spatial (S) and temporal (T) operators. The S-operator can be a graph convolutional network (GCN) (Kipf & Welling, 2017) or a Transformer (Vaswani et al., 2017). As for the T-operator, convolutional neural network (CNN), recurrent neural network (RNN) (Chung et al., 2014) or Transformer (Zeng et al., 2023) can be used.

**Temporal Graph Neural Network** A Graph Neural Network (GNN) is a type of neural network that operates on graph-structured data, such as social networks, citation networks, and molecular graphs. GNNs aim to learn node and graph representations by aggregating and transforming information from neighbouring nodes and edges (Wu et al., 2021b). GNNs have shown promising results in various applications, such as node classification, link prediction, and graph classification.

Temporal GNN (TGNN) (Longa et al., 2023) is an extension of GNNs which operates on temporal graphs $\mathcal{G}_t = (\boldsymbol{X}_t, \boldsymbol{A}_t)$ where $\boldsymbol{X}_t$ denotes the node features, and $\boldsymbol{A}_t$ is the evolving adjacency matrix. The TGNN architecture can be viewed as a neural network encoder-decoder pair $(f_\theta, g_\phi)$ (see Fig. 4).

A sequence of $m$ past graph snapshots is first encoded into an embedding $\boldsymbol{h}_t = f_\theta(\{\mathcal{G}_{t-m+1}, \cdots \mathcal{G}_t\})$, and then a sequence of $q$ future graph snapshots is estimated by the decoder as $\{\hat{\mathcal{G}}_{t+1}, \cdots \hat{\mathcal{G}}_{t+q}\} = g_\phi(\boldsymbol{h}_t)$. The parameters $(\theta, \phi)$ are trained to minimize the difference between the true sequence $\{\mathcal{G}_{t+1}, \cdots \mathcal{G}_{t+q}\}$ and the predicted sequence $\{\hat{\mathcal{G}}_{t+1}, \cdots \hat{\mathcal{G}}_{t+q}\}$. In node feature forecasting, the objective is to minimize the difference between the node feature matrices $\{\hat{\boldsymbol{X}}_{t+1}, \cdots \hat{\boldsymbol{X}}_{t+q}\}$ and $\{\boldsymbol{X}_{t+1}, \cdots \boldsymbol{X}_{t+q}\}$, while in temporal link prediction, the goal is to minimize the difference between the graph structures $\{\hat{\boldsymbol{A}}_{t+1}, \cdots \hat{\boldsymbol{A}}_{t+q}\}$ and $\{\boldsymbol{A}_{t+1}, \cdots \boldsymbol{A}_{t+q}\}$.

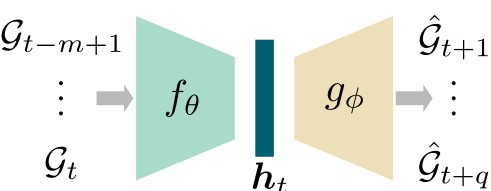

Figure 4: TGNN architecture.

There are two main approaches to implementing TGNNs: model evolution and embedding evolution. In *model evolution*, the parameters of a static GNN are updated over time to capture the temporal dynamics of the graph, e.g., `EvolveGCN` (Pareja et al., 2020). In *embedding evolution*, the GNN parameters remain fixed, and the node and edge embeddings are updated over time to learn the evolving graph structure and node features (Li et al., 2018; Zhao et al., 2019; Micheli & Tortorella, 2022; Wu et al., 2019; Fang et al., 2021; Liu et al., 2023). The TGNN methods are described in Appendix D.3.

**Linear Dynamical System** In a linear dynamical system (LDS) (Barber, 2012), the observation $\boldsymbol{y}_t$ is modelled as a linear function of the latent vector $\boldsymbol{h}_t$. The *transition model* dictates the temporal evolution of the latent state $\boldsymbol{h}_t = \mathbf{A}_t \boldsymbol{h}_{t-1} + \boldsymbol{\eta}_t$, with $\boldsymbol{\eta}_t \sim \mathcal{N}(\boldsymbol{\eta}; \bar{\boldsymbol{h}}_t, \boldsymbol{\Sigma}_t)$, and the *emission model* defines the relation between the observation and the latent state $\boldsymbol{y}_t = \mathbf{B}_t \boldsymbol{h}_t + \boldsymbol{\zeta}_t, \boldsymbol{\zeta}_t \sim \mathcal{N}(\boldsymbol{\zeta}_t; \bar{\boldsymbol{y}}_t, \boldsymbol{\Sigma}'_t)$. The LDS describes a first-order Markov model $p((\boldsymbol{y}_t, \boldsymbol{h}_t)_{t=1}^T) = p(\boldsymbol{h}_1)p(\boldsymbol{y}_1 \mid \boldsymbol{h}_1) \prod_{t=2}^T p(\boldsymbol{h}_t \mid \boldsymbol{h}_{t-1})p(\boldsymbol{y}_t \mid \boldsymbol{h}_t)$, where $p(\boldsymbol{h}_t \mid \boldsymbol{h}_{t-1}) = \mathcal{N}(\boldsymbol{h}_t; \mathbf{A}_t \boldsymbol{h}_{t-1} + \bar{\boldsymbol{h}}_t, \boldsymbol{\Sigma})$, and $p(\boldsymbol{y} - t \mid \boldsymbol{h}_t) = \mathcal{N}(\boldsymbol{y}_t; \mathbf{B}_t \boldsymbol{h}_t + \bar{\boldsymbol{y}}_t, \boldsymbol{\Sigma}'_t)$. Therefore a LDS is defined by the parameters $(\mathbf{A}_t, \mathbf{B}_t, \boldsymbol{\Sigma}_t, \boldsymbol{\Sigma}'_t, \bar{\boldsymbol{h}}_t, \bar{\boldsymbol{y}}_t)$ and initial state $\boldsymbol{h}_1$. In simplified models the parameters can be considered time-invariant. In the literature, LDS is also referred to as Kalman filter (Welch, 1997), or Gaussian state space model (Eleftheriadis et al., 2017).

**Gaussian Mixture Model** A Gaussian mixture model (GMM) (McLachlan et al., 2019) is a weighted sum of multiple Gaussian distribution components. An $M$-component GMM is defined as:

$$p(\boldsymbol{x}) = \sum_{i \in [M]} w_i \cdot \mathcal{N}(\boldsymbol{x}; \boldsymbol{\mu}_i, \boldsymbol{\Sigma}_i), \quad \sum_{i \in [M]} w_i = 1. \tag{4}$$

where $w_i$ denotes the probability of the sample belonging to the $i^{\text{th}}$ component. The parameters of the GMM $\{(w_i, \boldsymbol{\mu}_i, \boldsymbol{\Sigma}_i) : \forall i \in [M]\}$ are learnt through *expectation-maximisation* (EM) algorithm (Barber, 2012), *maximum a posteriori* (MAP) estimation, or *maximum likelihood* estimation (MLE) (Barber, 2012, Def. 8.30).

**Network Vector Autoregression** Network Vector Autoregression (NVAR) builds upon traditional vector autoregression models which capture the relationship among multiple time series by incorporating a network structure (Zhu et al., 2017). Developments, such as `Graph VARMA` (Isufi et al., 2019) and `Graph GARCH` (Hong et al., 2023) further refined the NVAR framework by addressing issues related to non-linear dependencies and heteroskedasticity. Although in this work, we have not compared our approach with NVAR methods, future work can be dedicated to comparing `mspace` with different variants of NVAR.

## 5 Results

**Baselines & Datasets**  We compare the performance of `mspace` with the following recent TGNN baselines: `DCRNN` (Li et al., 2018), `TGCN` (Zhao et al., 2019), `EGCN-H` (Pareja et al., 2020), `EGCN-O` (Pareja et al., 2020), `DynGESN` (Micheli & Tortorella, 2022), `GWNet` (Wu et al., 2019), `STGODE` (Fang et al., 2021), `FOGS` (Rao et al., 2022), `GRAM-ODE` (Liu et al., 2023), `LightCTS` (Lai et al., 2023). Additionally, we also evaluate the performance of classic autoregressive method `ARIMA` (Box & Pierce, 1970), and the famous LDS, the Kalman filter (Welch, 1997). We introduce two variants of the Kalman filter: `Kalman-`$x$, which considers the node features as observations, and `Kalman-`$\varepsilon$, which operates on the shocks. For more details, please see Appendix D.

Table 1: We use the **datasets** `tennis`, `wikimath`, `pedalme`, and `cpox` for single-step forecasting as they are relatively smaller in terms of number of nodes $n$ and samples $T$. For multi-step forecasting we use the larger *traffic* datasets `PEMS03`, `PEMS04`, `PEMS07`, `PEMS08`, `PEMSBAY`, and `METRLA`. The datasets `PEMS03/04/07/08` report traffic flow, while `PEMSBAY`, and `METRLA` report traffic speed.

|   | tennis | wikimath | pedalme | cpox | PEMS03 | PEMS04 | PEMS07 | PEMS08 | PEMSBAY | METRLA |
|---|---|---|---|---|---|---|---|---|---|---|
| $n$ | 1000 | 1068 | 15 | 20 | 358 | 307 | 883 | 170 | 325 | 207 |
| $T$ | 120 | 731 | 35 | 520 | 26K | 17K | 28K | 18K | 52K | 34K |

**Single-step Forecasting**  In Table 2, we have single-step forecasting RMSE results for various models with training ratio 0.9. The best result is marked **bold**, and the second-best is underlined.

The models `DCRNN`, `ECGN`, and `TGCN` exhibit similar performance across all datasets, which may be attributed to their use of convolutional GNNs for spatial encoding. `Kalman-`$\varepsilon$ performs poorly across all datasets, indicating challenges in establishing a state-space relation for shocks. In contrast, `Kalman-`$x$ performs notably well, outperforming other methods on `tennis` and `pedalme` datasets. We did not investigate why Kalman filters perform poorly when applied to shocks. However, it can be explored in future work.

For `wikimath` and `cpox`, `STGODE` shows the best performance, followed by `LightCTS` and `GRAM-ODE`, potentially due to a higher number of training samples.

Table 2: Single-step forecasting RMSE, $(M = 20)$.

|   | tennis | wikimath | pedalme | cpox |
|---|---|---|---|---|
| DynGESN | 150.41 | 906.85 | 1.25 | 0.95 |
| DCRNN | 155.43 | 1108.87 | 1.21 | 1.05 |
| EGCN-H | 155.44 | 1118.55 | 1.19 | 1.06 |
| EGCN-O | 155.43 | 1137.68 | 1.2 | 1.07 |
| TGCN | 155.43 | 1109.99 | 1.22 | 1.04 |
| LightCTS | 199.04 | 319.47 | 1.58 | 0.84 |
| GRAM-ODE | 206.50 | 484.90 | 0.99 | 0.98 |
| STGODE | 172.16 | **279.87** | 0.91 | **0.83** |
| mspace-S$\mu$ | 105.32 | 563.69 | 0.86 | 1.58 |
| mspace-S$\mathcal{N}$ | 117.23 | 725.42 | 1.35 | 2.11 |
| Kalman-$x$ | **73.01** | 792.6 | **0.66** | 1.42 |
| Kalman-$\varepsilon$ | 7.5K | 64K | 1.79 | 10.2 |

The light-weight methods such as `Kalman-`$x$ and `mspace` exploit the unavailability of enough training samples and perform better on `tennis` and `pedalme`.

We notice that `mspace-S`$\mu$ achieves a balanced performance between TGNN models and `Kalman-`$x$ across all datasets except for `cpox`. The subpar performance of `mspace-S*` on the `cpox` dataset may be attributed to the seasonal trend, given that it represents the weekly count of chickenpox cases.

**Multi-step Forecasting**  For the TGNN models, we use the $6:2:2$ train-validation-test chronological split in line with the experiments reported by the baselines. For `mspace` and `Kalman`, the train-test chronological split is $8:2$, as they do not require a validation set. In Table 3 we report the multi-step $q = 12$[2] forecasting RMSE, and mean absolute error (MAE) on the test set. For `mspace`, the queue size $M = 20$[3].

Figure 5 shows the RMSE of the models, normalized to the minimum RMSE for the dataset, plotted against the number of available training samples[4]. We observe that `mspace-T`$\mu$ performs competitively across all datasets with the exception of `METRLA`. Moreover, `mspace-T`$\mu$ demonstrates superior performance compared to `mspace-S`$\mu$ across all the datasets which suggests that temporal auto-correlation dominate spatial cross-correlation among the nodes.

---

[2] $q = 12$ corresponds to one hour in the traffic datasets used.
[3] a higher value of $M$ might give better estimates at the cost of higher memory usage and lower adaptability.
[4] We refer to the number of training samples as the training size.

Table 3: Multi-step forecasting RMSE and MAE, ($M = 20$).

| | PEMS03 | | PEMS04 | | PEMS07 | | PEMS08 | | PEMSBAY | | METRLA | |
|---|---|---|---|---|---|---|---|---|---|---|---|---|
| | RMSE | MAE | RMSE | MAE | RMSE | MAE | RMSE | MAE | RMSE | MAE | RMSE | MAE |
| GRAM-ODE | 26.40 | 15.72 | 31.05 | 19.55 | 34.42 | 21.75 | 25.17 | 16.05 | **3.34** | **1.67** | **6.64** | 3.44 |
| STGODE | 27.84 | 16.50 | 32.82 | 20.84 | 37.54 | 22.99 | 25.97 | 16.81 | 4.89 | 2.30 | 7.37 | 3.75 |
| DCRNN | 30.31 | 18.18 | 38.12 | 24.70 | 38.58 | 25.30 | 27.83 | 17.86 | 4.74 | 2.07 | 7.60 | 3.60 |
| ARIMA | 47.59 | 33.51 | 48.80 | 33.73 | 59.27 | 38.17 | 44.32 | 31.09 | 6.50 | 3.38 | 13.23 | 6.90 |
| GWNet | 32.94 | 19.85 | 39.70 | 25.45 | 42.78 | 26.85 | 31.05 | 19.13 | 4.85 | 1.95 | 7.81 | 3.53 |
| LightCTS | - | - | 30.14 | 18.79 | - | - | 23.49 | 14.63 | 4.32 | 1.89 | 7.21 | **3.42** |
| FOGS | **24.09** | **15.06** | 31.33 | 19.35 | **33.96** | **20.62** | 24.09 | 14.92 | - | - | - | - |
| mspace-S$\mu$ | 36.51 | 26.43 | 18.85 | 13.25 | 54.39 | 38.83 | 14.61 | 10.36 | 4.27 | 2.47 | 10.24 | 6.56 |
| mspace-T$\mu$ | 26.53 | 18.31 | **13.49** | **8.70** | 38.63 | 24.02 | **10.35** | **6.33** | 3.77 | 2.19 | 10.08 | 6.77 |
| Kalman-$x$ | 45.38 | 33.21 | 33.75 | 15.26 | 64.95 | 48.01 | 27.40 | 12.40 | 5.71 | 3.87 | 13.97 | 10.7 |
| Kalman-$\varepsilon$ | 749 | 619 | 818 | 709 | 2313 | 1988 | 460 | 399 | 50.2 | 43.1 | 127.1 | 109 |

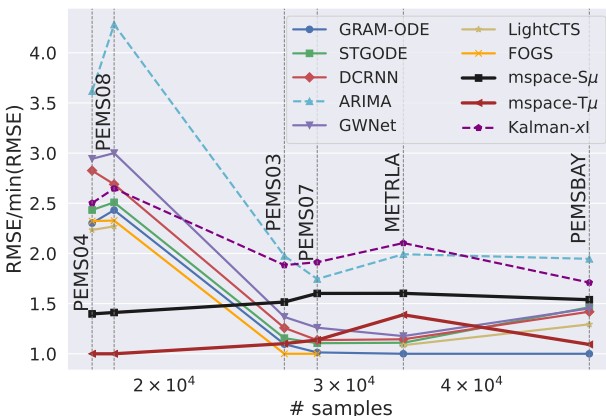

Figure 5: Multi-step forecasting normalized RMSE.

TGNN models, being neural networks, rely heavily on the amount of training data available. With the relatively small number of training samples in `PEMS04` and `PEMS08`, these models underperform. In contrast, both variants of `mspace` significantly surpass the state-of-the-art (SoTA), demonstrating their effectiveness with smaller datasets[5]. Furthermore, `mspace-T`$\mu$ ranks as the second-best model for the largest dataset, `PEMSBAY`. Therefore, we conclude that `mspace` offers consistent performance across datasets with varying sample sizes[6], and it is particularly advantageous when training data is limited.

In Fig. 6, we illustrate how the RMSE scales with the number of forecast steps $q$ for different variants of `mspace`[7]. The scaling law for `mspace-S*` appears linear, while for `mspace-T*`, it appears sublinear. We investigate this theoretically in Appendix A.

The TGNN baselines perform forecasting for $q = 12$ future steps, relying on the node features from the preceding 12 time steps as input. In contrast, `mspace` requires only the node features from the **two previous** time steps. Additionally, `mspace` has the flexibility to forecast for any $q \in \mathbb{N}$, whereas TGNN models are limited to forecasting up to the specified number of steps they were trained on. Moreover, `mspace` offers both probabilistic ($\Omega_{\mathcal{N}}$) and deterministic ($\Omega_{\mu}$) forecasts, a capability absent in the baselines. Finally, while TGNN baselines exploit the edge weights information for predictions, `mspace` achieves comparable results using only the graph structure.

---

[5]single-step forecasting datasets have prohibitively low number of samples ($< 800$), likely limiting `mspace`'s performance compared to multi-step forecasting with $17k+$ samples.

[6]Another approach to study the impact of training size is to use a fixed test set and vary the training fraction on a given dataset. While we did not run this experiment, it was suggested by a reviewer during the discussion and is worth mentioning.

[7]We acknowledge that including error scaling for the baselines could have provided a more comprehensive comparison.

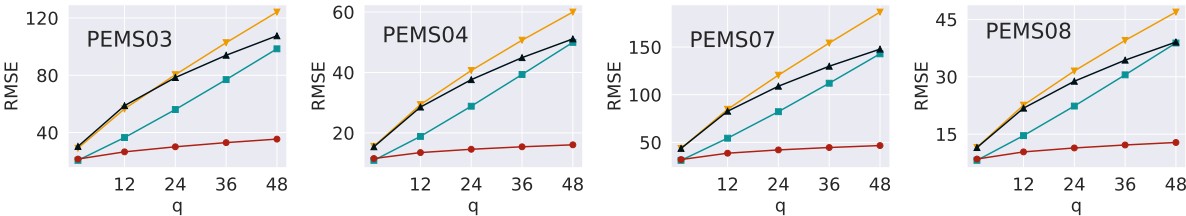

Figure 6: Scaling of error with the number of forecast steps $q$ using different `mspace` variants: ▼ `mspace-S`$\mathcal{N}$, ▲ `mspace-T`$\mathcal{N}$, ■ `mspace-S`$\mu$, ● `mspace-T`$\mu$.

## 6 Interpretability

In this section, we examine `mspace` in light of the following definition of Interpretability.

**Definition 6.1.** Consider data $\boldsymbol{x} \in \mathcal{D}$ which is processed by a model $\mathsf{F}_\theta$ to produce the output $\hat{\boldsymbol{y}} \in \mathcal{Y}$, i.e., $\hat{\boldsymbol{y}} = \mathsf{F}_\theta(\boldsymbol{x})$, where $\theta$ denotes the model parameters. Moreover, consider a true mapping $f : \boldsymbol{x} \mapsto \boldsymbol{y}$, $\forall \boldsymbol{x} \in \mathcal{D}$ where $\boldsymbol{y}$ is the ground truth associated with the input data $\boldsymbol{x}$. Then, an interpretable or explainable model $\mathsf{F}_\theta$ fulfils one or more of the following properties (Gilpin et al., 2018; Du et al., 2019):

- The internals of the model $\mathsf{F}_\theta$ can be explained in a way that is understandable to humans.

- The output $\hat{\boldsymbol{y}}$ can be explained in terms of the properties of the input $\boldsymbol{x}$, the input data distribution $\mathcal{D}$, and the model parameters $\theta$.

- The failure of a model on a given input data can be explained.

- For a certain distance metric $\Delta : \mathcal{Y} \times \mathcal{Y} \to \mathbb{R}^+$, theoretical bounds on the expected error $\mathbb{E}_{\boldsymbol{x} \sim \mathcal{D}}[\Delta(\boldsymbol{y}, \mathsf{F}_\theta(\boldsymbol{x}))]$ can be established based on the description of $\mathsf{F}_\theta$, supported by the assumptions on the input data distribution $\mathcal{D}$.

- It can be identified whether the model $\mathsf{F}_\theta$ is susceptible to training bias, and to what extent.

### 6.1 Explaining $\Psi_{\mathsf{S}}$

In Fig. 7, we depict two consecutive snapshots of a subgraph, focused on node $v$. The dashed circle highlights the corresponding 1-hop neighbourhood $\mathcal{U}_v$. At any time $t$, we draw green and red arrows next to the nodes to depict whether its node feature value increased or decreased, respectively.

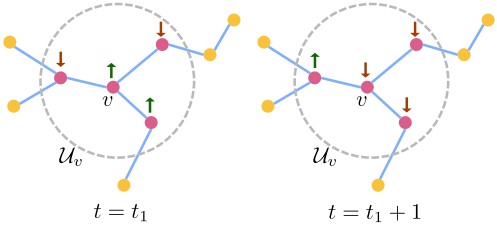

Figure 7: Consecutive subgraph snapshots.

The design of $\Psi_{\mathsf{S}}$ was inspired by the correlation dynamics of the stock market (Caraiani, 2014), where the inter-connectedness of various stocks exerts mutual influence on their respective prices. For instance, within the semiconductor sector, stocks such as `NVDA`, `AMD`, and `TSMC` often exhibit synchronised movements, with slight lead or lag. Similarly, the performance of gold mining stocks can offer insights into the future value of physical gold and companies engaged in precious metal trade. This concept transcends individual industries and encompasses competition across multiple sectors.

Let us record the states at two consecutive time-steps $\boldsymbol{s}_{t_1} = \begin{bmatrix} 1 & -1 & 1 & -1 \end{bmatrix}^\top$, and $\boldsymbol{s}_{t_1+1} = \begin{bmatrix} -1 & -1 & -1 & 1 \end{bmatrix}^\top$. At the state-level, we iterate through the time-steps, and collect all the states succeeding $\boldsymbol{s} = \begin{bmatrix} 1 & -1 & 1 & -1 \end{bmatrix}^\top$. If we then draw a random sample from this collection of succeeding states, we can predict whether the node feature value is more likely to *increase or decrease*. However, we are interested in predicting the *amount* of change. Therefore, at every time step when the state $\boldsymbol{s}_t$ matches $\boldsymbol{s} = \begin{bmatrix} 1 & -1 & 1 & -1 \end{bmatrix}^\top$, we collect the succeeding shock $\boldsymbol{\varepsilon}_{t+1}^{\langle \mathcal{U}_v \rangle}$ in a queue $\mathcal{Q}_v(\boldsymbol{s})$, i.e., at time $\tau$, $\mathcal{Q}_v(\boldsymbol{s}) = \left\{ \boldsymbol{\varepsilon}_{t+1}^{\langle \mathcal{U}_v \rangle} : \boldsymbol{s}_t = \boldsymbol{s}, \forall t < \tau \right\}$ with $|\mathcal{Q}_v(\boldsymbol{s})| \leq M$. The queue entries are then used to approximate a distribution from which a random sample is drawn during forecast.

In Fig. 8, we plot the normalized histogram of the trace $\mathrm{tr}(\cdot)$ of the covariance matrix $\boldsymbol{\Sigma}(\boldsymbol{s})$ of all the states $\boldsymbol{s} \in \mathcal{S}_v, v \in [n]$ for all the datasets used in multi-step forecasting. We notice that in both `PEMS04` and `PEMS08` the distribution of values is skewed to the left, with a concentration of data points at values close to zero . This explains the better-than-SoTA performance of `mspace-S`$\mu$ on these datasets. In contrast, the histogram of `METRLA` is completely away from zero, while for `PEMS03`, and `PEMS07` there are peaks near zero, but a major mass of the histogram is skewed away from zero. This explains the poor performance of `mspace-S`$\mu$ on these datasets.

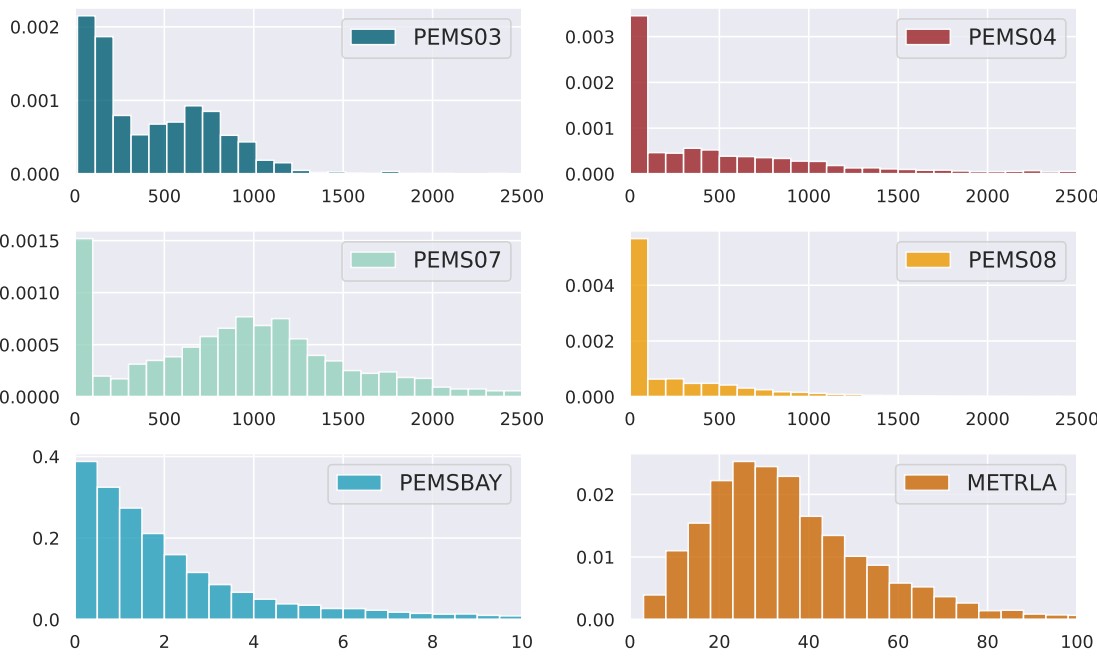

Figure 8: Normalized histogram of $\{\mathrm{tr}(\boldsymbol{\Sigma}(\boldsymbol{s})) : \forall \boldsymbol{s} \in \mathcal{S}_v, \forall v \in [n]\}$ for different datasets.

We represent data variance using the trace of the covariance matrix. Thus, if the variance histogram is close to zero, it indicates low variance. The error in estimating samples from a distribution is lower if the variance of the distribution is lower, and vice versa.

## 6.2 Explaining $\Psi_\mathrm{T}$

Next, we discuss the rationale behind $\Psi_\mathrm{T}$, which is designed to identify periodic patterns. For instance, in many traffic networks, trends exhibit weekly cycles, with distinct patterns on weekdays compared to weekends. Moreover, on an annual basis, the influence of holidays on traffic can be discerned, as people engage in shopping and other leisure activities. In Fig. 9, we have shown the traffic flow value of `PEMS04` with weekly (a) and daily (b) periodicity. For the weekly periodic view (a), the trend is more pronounced with less deviation from the mean while for the daily view (b), a scattered trend is visible with high variance across states.

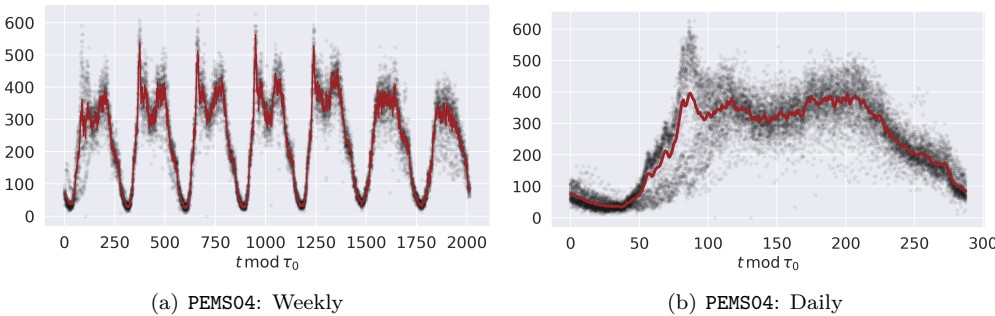

(a) PEMS04: Weekly

(b) PEMS04: Daily

Figure 9: Periodic trends in the traffic dataset PEMS04; the black points represent the data-points, and the red line is the mean estimate for each state $t \bmod \tau_0$.

## 6.3 Error Bounds

We present the error bounds of mspace in the following theorem, a detailed proof of which can be found in Appendix A.

**Theorem 6.1.** *The RMSE of mspace for a $q$-step node feature forecast is upper bounded as* $\mathrm{RMSE}(q) \leq \sqrt{\alpha q^2 + (3\alpha + \beta)q + (2\alpha + \beta)}$, *where* $\alpha, \beta \in \mathbb{R}^+$ *are constants that depend on the data, as well as the variant of the* mspace *algorithm.*

**Corollary 6.1.** *In the asymptotic case of large $q$, the RMSE grows linearly with $q$:* $\mathrm{RMSE}(q) = \mathcal{O}(q)$.

## 6.4 Complexity Analysis

We denote the *computational complexity* operator as $\mathfrak{C}(\cdot)$, and the *space complexity* operator as $\mathfrak{M}(\cdot)$, where the argument of each operator is an algorithm or a portion of an algorithm. The optional offline part of mspace is denoted by A, while the online part is denoted by B. In Table 4, we exhibit the computational and space complexities of the different mspace variants, where $b \triangleq \max_{v \in [n]} |\mathcal{U}_v|$ is the maximum degree. For more details please refer to Appendix B.

Table 4: Computational and space complexity of different mspace variants.

| | $\Psi_{\mathsf{S}}$ | $\Psi_{\mathsf{T}}$ |
|---|---|---|
| $\Omega_{\mathcal{N}}$ | $\mathfrak{C}(\mathsf{A}) = \mathcal{O}\left(ndb\left(rT + dbM\min\{rT, 2^{bd}\}\right)\right)$ $\mathfrak{C}(\mathsf{B}) = \mathcal{O}\left((1-r)Tnd^2b^2\left(qdb + M\min\left\{\frac{(1+r)}{2}T, 2^{bd}\right\}\right)\right)$ $\mathfrak{M}(\mathsf{A} \cup \mathsf{B}) = \mathcal{O}\left(db(M + db)\min\{T, 2^{bd}\}\right)$ | $\mathfrak{C}(\mathsf{A}) = \mathcal{O}\left(nrT + d^2Mn\tau_0\right)$ $\mathfrak{C}(\mathsf{B}) = \mathcal{O}\left((1-r)Tnd^2(qd + M\tau_0)\right)$ $\mathfrak{M}(\mathsf{A} \cup \mathsf{B}) = \mathcal{O}\left(d(M+d)\tau_0\right)$ |
| $\Omega_{\mu}$ | $\mathfrak{C}(\mathsf{A}) = \mathcal{O}\left(ndb\left(rT + M\min\{rT, 2^{bd}\}\right)\right)$ $\mathfrak{C}(\mathsf{B}) = \mathcal{O}\left((1-r)Tndb(q+M)\min\left\{\frac{(1+r)}{2}T, 2^{bd}\right\}\right)$ $\mathfrak{M}(\mathsf{A} \cup \mathsf{B}) = \mathcal{O}\left(Mdb\min\{T, 2^{bd}\}\right)$ | $\mathfrak{C}(\mathsf{A}) = \mathcal{O}\left(nrT + dMn\tau_0\right)$ $\mathfrak{C}(\mathsf{B}) = \mathcal{O}\left((1-r)Tnd(q+M)\tau_0\right)$ $\mathfrak{M}(\mathsf{A} \cup \mathsf{B}) = \mathcal{O}\left(Md\tau_0\right)$ |

**Theorem 6.2.** *For asymptotically large number of nodes $n$ and timesteps $T$, the computational complexity of* mspace *is* $\mathcal{O}(nT)$, *and the space complexity is* $\mathcal{O}(1)$ *across all variants.*

The proof is detailed in Appendix B.2.

## 7 Discussion

In this section we discuss the limitations of `mspace` and how they can be overcome. Firstly, `mspace` only considers binary edges, i.e.. $\boldsymbol{A} \in \{0,1\}^{n \times n}$ instead of a weighted adjacency matrix $\boldsymbol{A} \in \mathbb{R}^{n \times n}$. This does not imply that we have used datasets with binary edges, rather it means that we have used a binarized version of the adjacency matrix as input to `mspace` while the baselines exploited weighted edges. Secondly, we assume that the graph structure is fixed throughout, while for a truly dynamic graph, the graph structure should also be dynamic. Lastly, we have proposed two state functions: one that focuses on cross-correlation among the nodes, and the other that considers seasonality. Therefore, a state function which combines both can be studied in an extension of our work in the future.

**On creating a state function which combines $\Psi_{\mathtt{S}}$ and $\Psi_{\mathtt{T}}$** We can define $\Psi_{\mathtt{ST}} : \mathbb{R}^{|\mathcal{U}|d} \times \mathbb{N} \to \{-1,1\}^{|\mathcal{U}|d} \times \{0, 1, \cdots \tau_0 - 1\}$ as $\Psi_{\mathtt{ST}}\left(\boldsymbol{\varepsilon}^{\langle \mathcal{U}\rangle}, t\right) \triangleq \left[\mathrm{sign}(\boldsymbol{\varepsilon}^{\langle\mathcal{U}\rangle})^{\top} \quad t \bmod \tau_0\right]^{\top}$. In essence, the queues $\mathcal{Q}_v(\boldsymbol{s}), \forall \boldsymbol{s} \in \mathcal{S}_v, \forall v \in [n]$ in `mspace-ST` would have lesser entries compared to `mspace-S` which might lead to poor estimates and consequently make the algorithm data-intensive. Furthermore, in the step where we find the closest state $\boldsymbol{s}^*$, the spatial and temporal parts can be assigned different weights: $\boldsymbol{s}^* \leftarrow \arg\min_{\boldsymbol{s} \in \mathcal{S}_v} \left\| \left[\mathbf{1}_{d|\mathcal{U}_v|} \quad \gamma\right]^{\top} \odot \left(\boldsymbol{s} - \boldsymbol{s}_t^{\langle \mathcal{U}_v\rangle}\right)\right\|$, where $\gamma \in \mathbb{R}^+$.

**On benchmarking using diverse datasets** Experiments on more diverse datasets would help establish the performance of the proposed algorithm. In this work, we have used 4 non-traffic datasets for single-step forecasting, and 6 traffic datasets for multi-step. The proposed algorithm `mspace` has a general formulation, and is not designed specifically for traffic datasets; `mspace` can be applued to any graph whose node features (of any dimension) evolve with time. We also proposed a synthetic temporal graph generation method in Appendix C to alleviate the data scarcity issue in temporal graph learning.

## 8 Conclusion

In conclusion, our proposed algorithm, `mspace`, performs at par with the SoTA TGNN models across various spatio-temporal datasets. As an online learning algorithm, `mspace` is adaptive to changes in data distribution and is suitable for deployment in scenarios where training samples are limited. The interpretability of `mspace` sets it apart from black-box deep learning models, allowing for a clearer understanding of the underlying mechanisms driving predictions. This emphasis on interpretability represents a significant step forward in the field of temporal graph learning. In Sec. 7, we discussed the potential limitations of `mspace`.

In addition to the algorithm, we also introduce a synthetic temporal graph generator (see Appendix C) in which the features of the nodes evolve with the influence of their neighbours in a non-linear manner. These synthetic datasets can serve as a valuable resource for benchmarking algorithms.

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

## A   Error Bounds

**Upper Bound**   We derive the upper bound on the RMSE for $q$-step iterative forecast below.

*Proof of Theorem 6.1.* For nodes in $\mathcal{U}_v, v \in [n]$, the shock at time $t$ is sampled from a Gaussian distribution, the parameters of which depend on the previous shock $\hat{\varepsilon}_{t-1}^{\langle \mathcal{U}_v \rangle}$ through the state function:

$$\hat{\varepsilon}_t^{\langle \mathcal{U}_v \rangle} \sim \mathcal{N}\left(\hat{\varepsilon}; \boldsymbol{\mu}\left(\Psi_{\mathsf{S}}\left(\hat{\varepsilon}_{t-1}^{\langle \mathcal{U}_v \rangle}\right)\right), \boldsymbol{\Sigma}\left(\Psi_{\mathsf{S}}\left(\hat{\varepsilon}_{t-1}^{\langle \mathcal{U}_v \rangle}\right)\right)\right) \tag{5}$$

We denote the shock estimated for node $v$ at time $t$ as:

$$\hat{\varepsilon}_t(v) = \hat{\varepsilon}_t^{\langle \mathcal{U}_v \rangle}(v) \sim \mathcal{N}\left(\hat{\varepsilon}; \boldsymbol{\mu}_v\left(\Psi_{\mathsf{S}}\left(\hat{\varepsilon}_{t-1}^{\langle \mathcal{U}_v \rangle}\right)\right), \boldsymbol{\Sigma}_v\left(\Psi_{\mathsf{S}}\left(\hat{\varepsilon}_{t-1}^{\langle \mathcal{U}_v \rangle}\right)\right)\right) \tag{6}$$

The mean square error for $q$-step iterative node feature forecasting is defined as:

$$\mathrm{MSE}(q) \triangleq \frac{1}{ndq} \mathbb{E}\left[\sum_{v \in [n]} \sum_{i \in [q]} \left\|\sum_{j \in [i]} \hat{\varepsilon}_{t+j}(v) - \varepsilon_{t+j}(v)\right\|^2\right]$$

$$= \frac{1}{ndq} \sum_{v \in [n]} \sum_{i \in [q]} \mathbb{E}\left[\left\|\sum_{j \in [i]} \hat{\varepsilon}_{t+j}(v) - \varepsilon_{t+j}(v)\right\|^2\right]. \tag{7}$$

The shock difference between the true shock and predicted shock also follows a Gaussian distribution:

$$\hat{\varepsilon}_{t+j}(v) - \varepsilon_{t+j}(v) \sim \mathcal{N}\left(\varepsilon; \boldsymbol{\mu}_v\left(\Psi_{\mathsf{S}}\left(\hat{\varepsilon}_{t_j-1}^{\langle \mathcal{U}_v \rangle}\right)\right) - \varepsilon_{t+j}(v), \boldsymbol{\Sigma}_v\left(\Psi_{\mathsf{S}}\left(\hat{\varepsilon}_{t+j-1}^{\langle \mathcal{U}_v \rangle}\right)\right)\right). \tag{8}$$

Since, the sum of Gaussian r.v.s is also Gaussian, we have:

$$\sum_{j \in [i]} \hat{\varepsilon}_{t+j}(v) - \varepsilon_{t+j}(v) \sim \mathcal{N}\left(\varepsilon; \sum_{j \in [i]} \boldsymbol{\mu}_v\left(\Psi_{\mathsf{S}}\left(\hat{\varepsilon}_{t_j-1}^{\langle \mathcal{U}_v \rangle}\right)\right) - \varepsilon_{t+j}(v), \sum_{j \in [i]} \boldsymbol{\Sigma}_v\left(\Psi_{\mathsf{S}}\left(\hat{\varepsilon}_{t+j-1}^{\langle \mathcal{U}_v \rangle}\right)\right)\right). \tag{9}$$

Moreover, for a Gaussian r.v. $\mathbf{x} \sim \mathcal{N}(\mathbf{x}; \boldsymbol{\mu}, \boldsymbol{\Sigma})$, $\mathbb{E}\left[\|\boldsymbol{x}\|^2\right] = \|\boldsymbol{\mu}\|^2 + \mathrm{tr}(\boldsymbol{\Sigma})$.

$$\mathbb{E}\left[\left\|\sum_{j \in [i]} \hat{\varepsilon}_{t+j}(v) - \varepsilon_{t+j}(v)\right\|^2\right] = \left\|\sum_{j \in [i]} \boldsymbol{\mu}_v\left(\Psi_{\mathsf{S}}\left(\hat{\varepsilon}_{t+j-1}^{\langle \mathcal{U}_v \rangle}\right)\right) - \varepsilon_{t+j}(v)\right\|^2$$

$$+ \sum_{j \in [i]} \mathrm{tr}\left(\boldsymbol{\Sigma}_v\left(\Psi_{\mathsf{S}}\left(\hat{\varepsilon}_{t+j-1}^{\langle \mathcal{U}_v \rangle}\right)\right)\right). \tag{10}$$

$$\left\| \sum_{j \in [i]} \boldsymbol{\mu}_v \left( \Psi_{\mathtt{S}} \left( \hat{\varepsilon}_{t+j-1}^{\langle \mathcal{U}_v \rangle} \right) \right) - \boldsymbol{\varepsilon}_{t+j}(v) \right\| \leq \sum_{j \in [i]} \left\| \boldsymbol{\mu}_v \left( \Psi_{\mathtt{S}} \left( \hat{\varepsilon}_{t+j-1}^{\langle \mathcal{U}_v \rangle} \right) \right) - \boldsymbol{\varepsilon}_{t+j}(v) \right\|$$

$$\leq i \cdot \max_{j \in [i]} \left\| \boldsymbol{\mu}_v \left( \Psi_{\mathtt{S}} \left( \hat{\varepsilon}_{t+j-1}^{\langle \mathcal{U}_v \rangle} \right) \right) - \boldsymbol{\varepsilon}_{t+j}(v) \right\|$$

$$\leq i \cdot \max_{t,j \in \mathbb{N}} \left\| \boldsymbol{\mu}_v \left( \Psi_{\mathtt{S}} \left( \hat{\varepsilon}_{t+j-1}^{\langle \mathcal{U}_v \rangle} \right) \right) - \boldsymbol{\varepsilon}_{t+j}(v) \right\|$$

$$= i \cdot \sqrt{\alpha_{v,1}}. \tag{11}$$

$$\sum_{j \in [i]} \mathrm{tr} \left( \boldsymbol{\Sigma}_v \left( \Psi_{\mathtt{S}} \left( \hat{\varepsilon}_{t+j-1}^{\langle \mathcal{U}_v \rangle} \right) \right) \right) \leq i \cdot \max_{j \in [i]} \mathrm{tr} \left( \boldsymbol{\Sigma}_v \left( \Psi_{\mathtt{S}} \left( \hat{\varepsilon}_{t+j-1}^{\langle \mathcal{U}_v \rangle} \right) \right) \right) \leq i \cdot \alpha_{v,2}. \tag{12}$$

$$\mathbb{E} \left[ \left\| \sum_{j \in [i]} \hat{\varepsilon}_{t+j}(v) - \boldsymbol{\varepsilon}_{t+j}(v) \right\|^2 \right] \leq \alpha_{v,1} \cdot i^2 + \alpha_{v,2} \cdot i, \quad \alpha_{v,1}, \alpha_{v,2} \in \mathbb{R}^+. \tag{13}$$

$$\mathrm{MSE}(q) \leq \frac{1}{ndq} \sum_{v \in [n]} \sum_{i \in [q]} \alpha_{v,1} \cdot i^2 + \alpha_{v,2} \cdot i$$

$$= \frac{\sum_{v \in [n]} \alpha_{v,1}}{6nd}(q+1)(q+2) + \frac{\sum_{v \in [n]} \alpha_{v,2}}{2nd}(q+1). \tag{14}$$

Let $\alpha \triangleq \frac{1}{6nd} \sum_{v \in [n]} \alpha_{v,1}$, and $\beta \triangleq \frac{1}{2nd} \sum_{v \in [n]} \alpha_{v,2}$, then

$$\mathrm{MSE}(q) \leq \alpha q^2 + (3\alpha + \beta)q + (2\alpha + \beta). \tag{15}$$

By Jensen's inequality,

$$\mathrm{RMSE}(q) \leq \sqrt{\mathrm{MSE}(q)} \leq \sqrt{\alpha q^2 + (3\alpha + \beta)q + (2\alpha + \beta)}. \tag{16}$$

$$\square$$

The above proof is for `mspace-S`$\mathcal{N}$ and also applies to `mspace-T`$\mathcal{N}$. For `mspace-S`$\mu$ and `mspace-T`$\mu$, $\beta = 0$.

**Lower Bound** Similarly, we can find a lower bound on the MSE for $q$-step iterative forecast:

$$\mathbb{E} \left[ \left\| \sum_{j \in [i]} \hat{\varepsilon}_{t+j}(v) - \boldsymbol{\varepsilon}_{t+j}(v) \right\|^2 \right] \geq \sum_{j \in [i]} \mathrm{tr} \left( \boldsymbol{\Sigma}_v \left( \Psi_{\mathtt{S}} \left( \hat{\varepsilon}_{t+j-1}^{\langle \mathcal{U}_v \rangle} \right) \right) \right)$$

$$\geq i \cdot \min_{j \in [i]} \mathrm{tr} \left( \boldsymbol{\Sigma}_v \left( \Psi_{\mathtt{S}} \left( \hat{\varepsilon}_{t+j-1}^{\langle \mathcal{U}_v \rangle} \right) \right) \right) = i \cdot \alpha_{v,3}. \tag{17}$$

$$\mathrm{MSE}(q) \geq \frac{1}{ndq} \sum_{v \in [n]} \sum_{i \in [q]} i \cdot \alpha_{v,3} = \Big( \underbrace{\frac{1}{nd} \sum_{v \in [n]} \alpha_{v,3}}_{\triangleq \beta'} \Big) \cdot (q+1) = \beta' q + \beta'. \tag{18}$$

# B Complexity Analysis

## B.1 Computational Complexity

We denote the computational complexity operator as $\mathfrak{C}(\cdot)$, the argument of which is an algorithm or part of an algorithm. The optional offline part of the algorithm is denoted as A while the online part is denoted as B.

---

**Algorithm 2** mspace-S$\mathcal{N}$

---

**Input** $\mathcal{G} = (\mathcal{V}, \mathcal{E}, \boldsymbol{X})$, $r \in [0, 1)$, $q, M$
**Output** $\hat{\varepsilon}_t(v)$, $\quad \forall v \in \mathcal{V}, t \in [\lfloor r \cdot T \rfloor, T]$

1: $\boldsymbol{\varepsilon}_t(v) \leftarrow \boldsymbol{x}_t(v) - \boldsymbol{x}_{t-1}(v)$, $\quad \forall v \in \mathcal{V}, t \in [T]$
    *Offline training* (A)*:*
2: **for** $t \in [\lfloor r \cdot T \rfloor]$ **do**
3:     **for** $v \in \mathcal{V}$ **do**
4:         $\boldsymbol{s}_t^{\langle \mathcal{U}_v \rangle} \leftarrow \Psi\left(\boldsymbol{\varepsilon}_t^{\langle \mathcal{U}_v \rangle}\right)$                       $\triangleright \sum_{v \in \mathcal{V}} d|\mathcal{U}_v|$
5:         $\mathcal{S}_v \leftarrow \mathcal{S}_v \cup \left\{\boldsymbol{s}_t^{\langle \mathcal{U}_v \rangle}\right\}$                       $\triangleright n$
6:         $\mathcal{Q}_v\left(\boldsymbol{s}_t^{\langle \mathcal{U}_v \rangle}\right) \leftarrow$ enqueue $\boldsymbol{\varepsilon}_{t+1}^{\langle \mathcal{U}_v \rangle}$             $\triangleright n$
7:     **end for**
8: **end for**
9: $\boldsymbol{\mu}_v(\boldsymbol{s}) \leftarrow \text{mean}(\mathcal{Q}_v(\boldsymbol{s}))$, $\quad \forall s \in \mathcal{S}_v, v \in \mathcal{V}$          $\triangleright \sum_{v \in \mathcal{V}} d|\mathcal{U}_v||\mathcal{S}_v|M$
10: $\boldsymbol{\Sigma}_v(\boldsymbol{s}) \leftarrow \text{covariance}(\mathcal{Q}_v(\boldsymbol{s}))$, $\quad \forall s \in \mathcal{S}_v, v \in \mathcal{V}$    $\triangleright \sum_{v \in \mathcal{V}} (d|\mathcal{U}_v|)^2|\mathcal{S}_v|M$
    *Online learning* (B)*:*
11: **for** $t \in [\lfloor r \cdot T \rfloor, T - q]$ **do**
12:     **for** $v \in \mathcal{V}$ **do**
13:         $\boldsymbol{s}_t^{\langle \mathcal{U}_v \rangle} \leftarrow \Psi\left(\boldsymbol{\varepsilon}_t^{\langle \mathcal{U}_v \rangle}\right)$                      $\triangleright \sum_{v \in \mathcal{V}} d|\mathcal{U}_v|$
14:         $\boldsymbol{s}^* \leftarrow \arg\min_{\boldsymbol{s} \in \mathcal{S}_v} \left\| \boldsymbol{s} - \boldsymbol{s}_t^{\langle \mathcal{U}_v \rangle} \right\|$           $\triangleright \sum_{v \in \mathcal{V}} d|\mathcal{U}_v||\mathcal{S}_v|$
15:         $\hat{\boldsymbol{\varepsilon}}_{t+1}^{\langle \mathcal{U}_v \rangle} \sim \mathcal{N}(\boldsymbol{\varepsilon}; \boldsymbol{\mu}_v(\boldsymbol{s}^*), \boldsymbol{\Sigma}_v(\boldsymbol{s}^*))$         $\triangleright \sum_{v \in \mathcal{V}} (|\mathcal{U}_v|d)^3$
16:         **for** $k \in [2, q]$ **do**
17:             $\boldsymbol{s}^* \leftarrow \arg\min_{\boldsymbol{s} \in \mathcal{S}_v} \left\| \boldsymbol{s} - \Psi\left(\hat{\boldsymbol{\varepsilon}}_{t+k-1}^{\langle \mathcal{U}_v \rangle}\right) \right\|$    $\triangleright (q - 1) \times \sum_{v \in \mathcal{V}} d|\mathcal{U}_v|(1 + |\mathcal{S}_v|)$
18:             $\hat{\boldsymbol{\varepsilon}}_{t+k}^{\langle \mathcal{U}_v \rangle} \sim \mathcal{N}(\boldsymbol{\varepsilon}; \boldsymbol{\mu}_v(\boldsymbol{s}^*), \boldsymbol{\Sigma}_v(\boldsymbol{s}^*))$         $\triangleright (q - 1) \times \sum_{v \in \mathcal{V}} (|\mathcal{U}_v|d)^3$
19:         **end for**
20:         $\hat{\varepsilon}_{t+k}(v) \leftarrow \hat{\varepsilon}_{t+k}^{\langle \mathcal{U}_v \rangle}(v)$, $\quad \forall k \in [q]$
21:         Update $\mathcal{S}_v, \mathcal{Q}_v$                          $\triangleright 2n$
22:         Update $\boldsymbol{\mu}_v(\boldsymbol{s}), \boldsymbol{\Sigma}_v(\boldsymbol{s})$, $\quad \forall \boldsymbol{s} \in \mathcal{S}_v$    $\triangleright \sum_{v \in \mathcal{V}} (d|\mathcal{U}_v| + d^2|\mathcal{U}_v|^2)|\mathcal{S}_v|M$
23:     **end for**
24: **end for**

---

Computational complexity of offline training for mspace-S$\mathcal{N}$ can be written as:

$$\mathfrak{C}(\mathsf{A}) = \mathcal{O}\Bigg( \underbrace{\lfloor rT \rfloor d \sum_v |\mathcal{U}_v|}_{[4]} + \underbrace{\lfloor rT \rfloor 2n}_{[5],[6]} + \underbrace{dM \sum_v |\mathcal{U}_v||\mathcal{S}_v|}_{[9](\text{mean})} + \underbrace{d^2 M \sum_v |\mathcal{U}_v|^2|\mathcal{S}_v|}_{[10](\text{covariance})} \Bigg). \tag{19}$$

Computational complexity of online learning for `mspace-S`$\mathcal{N}$ can be written as:

$$\mathfrak{C}(\mathsf{B}) = \mathcal{O}\left(\sum_{t=\lceil rT \rceil}^{T-q} \left\{ \underbrace{dq\sum_v |\mathcal{U}_v|}_{[13],[17]} + \underbrace{dq\sum_v |\mathcal{U}_v||\mathcal{S}_v|}_{[14],[17]} + \underbrace{d^3 q \sum_v |\mathcal{U}_v|^3}_{[15],[18](\text{sampling})} + \underbrace{2n}_{[21]} \right.\right.$$
$$\left.\left. + \underbrace{dM\sum_v |\mathcal{U}_v||\mathcal{S}_v|}_{[22](\text{mean})} + \underbrace{d^2 M \sum_v |\mathcal{U}_v|^2 |\mathcal{S}_v|}_{[22](\text{covariance})} \right\}\right). \qquad (20)$$

**Lemma B.1.** *The computational complexity of* `mspace-S`$\mathcal{N}$ *is:*

$$\mathfrak{C}(\mathsf{A}) = \mathcal{O}\left( dbnrT + d^2 b^2 Mn \cdot \min\{rT, 2^{bd}\} \right),$$
$$\mathfrak{C}(\mathsf{B}) = \mathcal{O}\left( (1-r)Tnd^2 b^2 \left( qdb + M \cdot \min\left\{ \frac{(1+r)}{2}T, 2^{bd} \right\} \right) \right),$$

*where* $b = \max_{v \in [n]} |\mathcal{U}_v|$.

*Proof.* We denote the maximum degree of a node as $b \triangleq \max_{v \in [n]} |\mathcal{U}_v| < n$ which does not necessarily scale with $n$ unless specified by the graph definition. Furthermore, the total number of states observed for a node till time step $t \in \mathbb{N}$ cannot exceed $t$, i.e., $|\mathcal{S}_v| \leq t$. We also know the total number of states theoretically possible for node $v$ is $2^{|\mathcal{U}_v|d}$ for $\Psi_{\mathsf{S}}(\cdot)$. Therefore, the number of states observed till time $t$ for node $v$ is upper bounded as: $|\mathcal{S}_v| \leq \min\{t, 2^{bd}\}$. Based on this, we can simplify equation 19, and equation 20 as follows:

$$\mathfrak{C}(\mathsf{A}) = \mathcal{O}\left( dbnrT + 2nrT + (dbM + d^2 b^2 M) \cdot n\min\{rT, 2^{bd}\} \right)$$
$$= \mathcal{O}\left( dbnrT + d^2 b^2 Mn \cdot \min\{rT, 2^{bd}\} \right).$$
$$\mathfrak{C}(\mathsf{B}) = \mathcal{O}\left( \sum_{t=\lceil rT \rceil}^{T-q} qdbn + qd^3 b^3 n + 2n + db(q+M)n \cdot \min\{t, 2^{bd}\} + d^2 b^2 Mn \cdot \min\{t, 2^{bd}\} \right)$$
$$= \mathcal{O}\left( \sum_{t=\lceil rT \rceil}^{T-q} qd^3 b^3 n + (db(q+M) + d^2 b^2 M)n \cdot \min\{t, 2^{bd}\} \right)$$
$$= \mathcal{O}\left( (1-r)T \cdot qd^3 b^3 n + d^2 b^2 Mn \cdot \min\{(1-r^2)T^2, 2^{bd}(1-r)T\} \right)$$
$$= \mathcal{O}\left( (1-r)Tn \left( qd^3 b^3 + d^2 b^2 M \cdot \min\left\{ \frac{(1+r)}{2}T, 2^{bd} \right\} \right) \right).$$

$\square$

**Lemma B.2.** *The computational complexity of* `mspace-S`$\mu$ *is:*

$$\mathfrak{C}(\mathsf{A}) = \mathcal{O}\left( dbnrT + dbMn \cdot \min\{rT, 2^{bd}\} \right),$$
$$\mathfrak{C}(\mathsf{B}) = \mathcal{O}\left( (1-r)Tndb(q+M) \cdot \min\left\{ \frac{(1+r)}{2}T, 2^{bd} \right\} \right).$$

*Proof.* The sampling steps [15], and [18] in Algorithm 2 are replaced with $\hat{\varepsilon}_t^{\langle \mathcal{U}_v \rangle} \leftarrow \boldsymbol{\mu}(\boldsymbol{s}^*)$ which has a computational complexity of $\mathcal{O}(d|\mathcal{U}_v|)$. Moreover, $\Omega_\mu(\cdot)$ does not require the covariance matrix, therefore we

do not need to compute it. We simplify the computational complexity expressions as:

$$\mathfrak{C}(\mathsf{A}) = \mathcal{O}\left(\lfloor rT \rfloor d \sum_v |\mathcal{U}_v| + \lfloor rT \rfloor 2n + dM \sum_v |\mathcal{U}_v||\mathcal{S}_v|\right)$$
$$= \mathcal{O}\left(dbnrT + dbMn \cdot \min\{rT, 2^{bd}\}\right).$$
$$\mathfrak{C}(\mathsf{B}) = \mathcal{O}\left(\sum_{t=\lceil rT \rceil}^{T-q} \left\{ dq \sum_v |\mathcal{U}_v| + dq \sum_v |\mathcal{U}_v||\mathcal{S}_v| + \underbrace{dq \sum_v |\mathcal{U}_v|}_{(\text{sampling})} + 2n + dM \sum_v |\mathcal{U}_v||\mathcal{S}_v|\right\}\right)$$
$$= \mathcal{O}\left(\sum_{t=\lceil rT \rceil}^{T-q} 2qdbn + 2n + db(q+M)n \cdot \min\{t, 2^{bd}\}\right)$$
$$= \mathcal{O}\left((1-r)Tndb(q+M) \cdot \min\left\{\frac{(1+r)}{2}T, 2^{bd}\right\}\right).$$

$\square$

**Lemma B.3.** *The computational complexity of* `mspace-T𝒩` *is:*

$$\mathfrak{C}(\mathsf{A}) = \mathcal{O}\left(nrT + d^2 Mn\tau_0\right),$$
$$\mathfrak{C}(\mathsf{B}) = \mathcal{O}\left((1-r)Tnd^2 \cdot (M\tau_0 + qd)\right).$$

*Proof.* For the state function $\Psi_{\mathsf{T}}$, the total number of states for any node is the period $\tau_0 \in \mathbb{N}$, i.e., $|\mathcal{S}_v| \leq \tau_0$. Moreover, the state calculation $s_t \leftarrow \Psi(t)$ has computational complexity of $\mathcal{O}(1)$. Most importantly, for $\Psi_{\mathsf{T}}$, $b = 1$ as it only focuses on the seasonal trends.

$$\mathfrak{C}(\mathsf{A}) = \mathcal{O}\left(\lfloor rT \rfloor \sum_v 1 + \lfloor rT \rfloor 2n + dM \sum_v |\mathcal{U}_v||\mathcal{S}_v| + d^2 M \sum_v |\mathcal{U}_v|^2 |\mathcal{S}_v|\right)$$
$$= \mathcal{O}\left(3nrT + dMn\tau_0 + d^2 Mn\tau_0\right) = \mathcal{O}\left(nrT + d^2 Mn\tau_0\right).$$
$$\mathfrak{C}(\mathsf{B}) = \mathcal{O}\left(\sum_{t=\lceil rT \rceil}^{T-q} \left\{ q \sum_v 1 + dq \sum_v |\mathcal{U}_v||\mathcal{S}_v| + d^3 q \sum_v |\mathcal{U}_v|^3 + 2n\right.\right.$$
$$\left.\left. + dM \sum_v |\mathcal{U}_v||\mathcal{S}_v| + d^2 M \sum_v |\mathcal{U}_v|^2 |\mathcal{S}_v|\right\}\right)$$
$$= \mathcal{O}\left(\{q + dq\tau_0 + qd^3 + 2 + dM\tau_0 + d^2 M\tau_0\} \cdot n(1-r)T\right)$$
$$= \mathcal{O}\left((1-r)Tnd^2 \cdot (M\tau_0 + qd)\right).$$

$\square$

**Lemma B.4.** *The computational complexity of* `mspace-Tμ` *is:*

$$\mathfrak{C}(\mathsf{A}) = \mathcal{O}\left(nrT + dMn\tau_0\right),$$
$$\mathfrak{C}(\mathsf{B}) = \mathcal{O}\left((1-r)Tn \cdot d(q+M)\tau_0\right).$$

*Proof.* Based on the explanation provided for `mspace-T`$\mathcal{N}$, we simplify the computational complexity expressions for `mspace-T`$\mu$ as:

$$
\begin{aligned}
\mathfrak{C}(\mathsf{A}) &= \mathcal{O}\left(\lfloor rT \rfloor \sum_v 1 + \lfloor rT \rfloor 2n + dM \sum_v |\mathcal{S}_v|\right) \\
&= \mathcal{O}\left(3nrT + dMn\tau_0\right) = \mathcal{O}\left(nrT + dMn\tau_0\right). \\
\mathfrak{C}(\mathsf{B}) &= \mathcal{O}\left(\sum_{t=\lceil rT \rceil}^{T-q} \left\{q\sum_v 1 + dq\sum_v |\mathcal{S}_v| + 2n + dM\sum_v |\mathcal{S}_v|\right\}\right) \\
&= \mathcal{O}\left(\{q + dq\tau_0 + 2 + dM\tau_0\} \cdot n(1-r)T\right) = \mathcal{O}\left((1-r)Tn \cdot d(q+M)\tau_0\right).
\end{aligned}
$$

$\square$

## B.2 Space Complexity

We denote the space complexity operator as $\mathfrak{M}(\cdot)$, the argument of which is an algorithm or part of an algorithm. The variables in offline training $\mathsf{A}$ are re-used in online learning $\mathsf{B}$. Therefore, we can say that $\mathfrak{M}(\mathsf{B}) = \mathfrak{M}(\mathsf{A} \cup \mathsf{B})$.

In an implementation of `mspace` where forecasting is sequentially performed for each node $v \in [n]$, memory space can be efficiently reused, except for storing the outputs. This approach optimises memory usage, resulting in a space complexity characterised by:

$$
\mathfrak{M}(\mathsf{A} \cup \mathsf{B}) = \mathcal{O}\left(\max_{\substack{v \in [n], \\ t \in [T]}} \underbrace{d|\mathcal{U}_v||\mathcal{S}_v|}_{\mathcal{S}_v} + \underbrace{cMd|\mathcal{U}_v||\mathcal{S}_v|}_{\mathcal{Q}_v(\boldsymbol{s})\,\forall \boldsymbol{s} \in \mathcal{S}_v} + \underbrace{cd|\mathcal{U}_v||\mathcal{S}_v|}_{\boldsymbol{\mu}_v(\boldsymbol{s})\,\forall \boldsymbol{s} \in \mathcal{S}_v} + \underbrace{c(d|\mathcal{U}_v|)^2|\mathcal{S}_v|}_{\boldsymbol{\Sigma}_v(\boldsymbol{s})\,\forall \boldsymbol{s} \in \mathcal{S}_v} + \underbrace{d|\mathcal{U}_v|}_{\boldsymbol{s}^*}\right). \tag{21}
$$

**Lemma B.5.** *The space complexity of `mspace-S`$\mathcal{N}$ is $\mathfrak{M}(\mathsf{A} \cup \mathsf{B}) = \mathcal{O}\left(db(M+db) \cdot \min\{T, 2^{bd}\}\right)$.*

*Proof.* Simplifying equation 21 results in:

$$
\begin{aligned}
\mathfrak{M}(\mathsf{A} \cup \mathsf{B}) &= \mathcal{O}\left(\max_{\substack{v \in [n], \\ t \in [T]}} (db + cMdb + cdb + cd^2b^2)|\mathcal{S}_v| + db\right) \\
&= \mathcal{O}\left((cMdb + cd^2b^2) \cdot \max_{t \in [T]} \min\{t, 2^{bd}\}\right) = \mathcal{O}\left(db(M+db) \cdot \min\{T, 2^{bd}\}\right).
\end{aligned}
$$

$\square$

**Lemma B.6.** *The space complexity of `mspace-S`$\mu$ is $\mathfrak{M}(\mathsf{A} \cup \mathsf{B}) = \mathcal{O}\left(Mdb \cdot \min\{T, 2^{bd}\}\right)$.*

*Proof.* Some space is saved in `mspace-S`$\mu$, as we do not need to store the covariance matrices.

$$
\mathfrak{M}(\mathsf{A} \cup \mathsf{B}) = \mathcal{O}\left(\max_{\substack{v \in [n], \\ t \in [T]}} (db + cMdb + cdb)|\mathcal{S}_v| + db\right) = \mathcal{O}\left(Mdb \cdot \min\{T, 2^{bd}\}\right).
$$

$\square$

**Lemma B.7.** *The space complexity of `mspace-T`$\mathcal{N}$ is $\mathfrak{M}(\mathsf{A} \cup \mathsf{B}) = \mathcal{O}\left(d(M+d)\tau_0\right)$.*

*Proof.* As explained earlier, for the state function $\Psi_{\mathtt{T}}$, $b = 1$. Therefore, the queues only store the shock vectors for a single node, and not the neighbours. The space complexity expression is simplified as:

$$\mathfrak{M}(\mathsf{A} \cup \mathsf{B}) = \mathcal{O}\left( \max_{\substack{v \in [n], \\ t \in [T]}} (d + cMd + cd + cd^2)|\mathcal{S}_v| + db \right) = \mathcal{O}\Big( d(M + d)\tau_0 \Big).$$

$\square$

**Lemma B.8.** *The space complexity of* `mspace-T`$\mu$ *is* $\mathcal{O}\Big( Md\tau_0 \Big)$.

*Proof.* $\mathfrak{M}(\mathsf{A} \cup \mathsf{B}) = \mathcal{O}\left( \max_{\substack{v \in [n], \\ t \in [T]}} (d + cMd + cd)|\mathcal{S}_v| + d \right) = \mathcal{O}\Big( Md\tau_0 \Big).$ $\square$

**Asymptotic Analysis**  Theorem 6.2 states that *for asymptotically large number of nodes $n$ and timesteps $T$, the computational complexity of* `mspace` *is* $\mathcal{O}(nT)$*, and the space complexity is* $\mathcal{O}(1)$ *across all variants.*

*Proof.* We analyse the lemmas B.1-B.8 introduced in this section for the asymptotic case of very large $n$ and $T$. For very large $T$, $\min\left\{ \frac{(1+r)}{2}T, 2^{bd} \right\} \to 2^{bd}$. Similarly, $\min\{T, 2^{bd}\} \to 2^{bd}$. Considering the terms $r, d, M, q, \tau_0, b$ as constants, the computational complexity for both offline and online parts of all the `mspace` variants becomes $\mathcal{O}(nT)$ for asymptotically large $n, T$.

Furthermore, the space complexity terms lack $n$ or $T$ for very large $T$, which allows us to conclude that the space complexity of all the variants of `mspace` is constant, i.e., $\mathcal{O}(1)$. $\square$

## C  Synthetic Datasets & Experiments

In traffic datasets, seasonality outweighs cross-nodal correlation, making it challenging to assess the efficacy of a TGL algorithms on node feature forecasting task. To address this gap, we propose a synthetic dataset generation technique in line with the design idea of `mspace` which is described in Algorithm 3.

---

**Algorithm 3** Synthetic Data Generation

**Input** $\mathcal{G} = (\mathcal{V}, \mathcal{E})$, $d$, $\mu_{\min}$, $\mu_{\max}$, $\sigma_{\min}^2$, $\sigma_{\max}^2$, $\mu_0$, $\sigma_0^2$, $\tau$, $\mu_\tau$, $\sigma_\tau^2$.

1:  $\boldsymbol{\varepsilon}_0 \sim \text{Bernoulli}^{nd}\left(\frac{1}{2}\right)$
2:  $\boldsymbol{x}_0 \sim \mathcal{N}(\boldsymbol{x}; \mu_0\boldsymbol{1}, \sigma_0^2\boldsymbol{I})$
3:  **for** $t \in [T]$ **do**
4:  $\quad \boldsymbol{s}_{t-1} \leftarrow \Psi_{\mathtt{S}}(\boldsymbol{\varepsilon}_{t-1})$
5:  $\quad$ **if** $\boldsymbol{s}_{t-1} \notin \mathcal{S}$ **then**
6:  $\quad\quad \mathcal{S} \leftarrow \mathcal{S} \cup \{\boldsymbol{s}_{t-1}\}$
7:  $\quad\quad \boldsymbol{\mu}(\boldsymbol{s}_{t-1}) \sim \text{Uniform}^{nd}(\mu_{\min}, \mu_{\max})$
8:  $\quad\quad \tilde{\boldsymbol{\Sigma}} \sim \text{Uniform}^{nd \times nd}(\sigma_{\min}^2, \sigma_{\max}^2)$
9:  $\quad\quad \hat{\boldsymbol{\Sigma}} \leftarrow \frac{1}{2}\left( \tilde{\boldsymbol{\Sigma}} + \tilde{\boldsymbol{\Sigma}}^\top \right)$
10: $\quad\quad \boldsymbol{\Sigma}(\boldsymbol{s}_{t-1}) \leftarrow \hat{\boldsymbol{\Sigma}} \odot (\boldsymbol{A} \otimes \boldsymbol{1}_{d \times d})$
11: $\quad$ **end if**
12: $\quad \boldsymbol{\varepsilon}_t \sim \mathcal{N}(\boldsymbol{\varepsilon}; \boldsymbol{\mu}(\boldsymbol{s}_{t-1}), \boldsymbol{\Sigma}(\boldsymbol{s}_{t-1}))$
13: $\quad \boldsymbol{x}_t = \boldsymbol{x}_{t-1} + \boldsymbol{\varepsilon}_t$
14: **end for**
15: **if** $\tau > 0$ **then**
16: $\quad \boldsymbol{y}_t \sim \mathcal{N}(\boldsymbol{y}; \mu_\tau\boldsymbol{1}, \sigma_\tau^2\boldsymbol{I}) \quad \forall t \in [\tau]$
17: $\quad \boldsymbol{x}_t \leftarrow \boldsymbol{x}_t + \boldsymbol{y}_{t \bmod \tau} \quad \forall t \in [T]$
18: **end if**

---

In steps 8-10, we construct a covariance matrix adhering to Assumption 2.2, and in step 12, we sample the shock from a multivariate normal distribution. In steps 16-17, a random signal $\boldsymbol{y}$ is tiled with period $\tau$ and added to the node features to introduce seasonality into the dataset.

The synthetic datasets can be utilized to analyze how various factors such as graph structure, periodicity, connectivity, sample size, and other parameters affect error metrics.

We generate datasets through Algorithm 3 by supplying the parameters outlined in Table 5. For each dataset, we create multiple random instances and report the mean and standard deviation of the metrics in the results.

Table 5: Parameters for different synthetic dataset packages.

| Dataset | $\mathcal{G} \sim$ | $d$ | $T$ | $\mu_{\min}$ | $\mu_{\max}$ | $\sigma_{\min}$ | $\sigma_{\max}$ | $\mu_0$ | $\sigma_0$ | $\tau$ | $\mu_\tau$ | $\sigma_\tau$ |
|---|---|---|---|---|---|---|---|---|---|---|---|---|
| SYN01 | $\mathfrak{G}_{\mathrm{ER}}(20, 0.2)$ | 1 | $10^3$ | $-200$ | 200 | 40 | 50 | $2 \times 10^4$ | 5000 | 100 | 100 | 20 |
| SYN02 | $\mathfrak{G}_{\mathrm{ER}}(20, 0.2)$ | 1 | $10^3$ | $-200$ | 200 | 40 | 50 | $2 \times 10^4$ | 5000 | 0 | | |
| SYN03 | $\mathfrak{G}_{\mathrm{ER}}(40, 0.5)$ | 1 | $10^3$ | $-400$ | 400 | 30 | 40 | $10^4$ | 2000 | 0 | | |
| SYN04 | $\mathfrak{G}_{\mathrm{ER}}(40, 0.5)$ | 1 | $10^4$ | $-400$ | 400 | 30 | 40 | $10^4$ | 2000 | 0 | | |

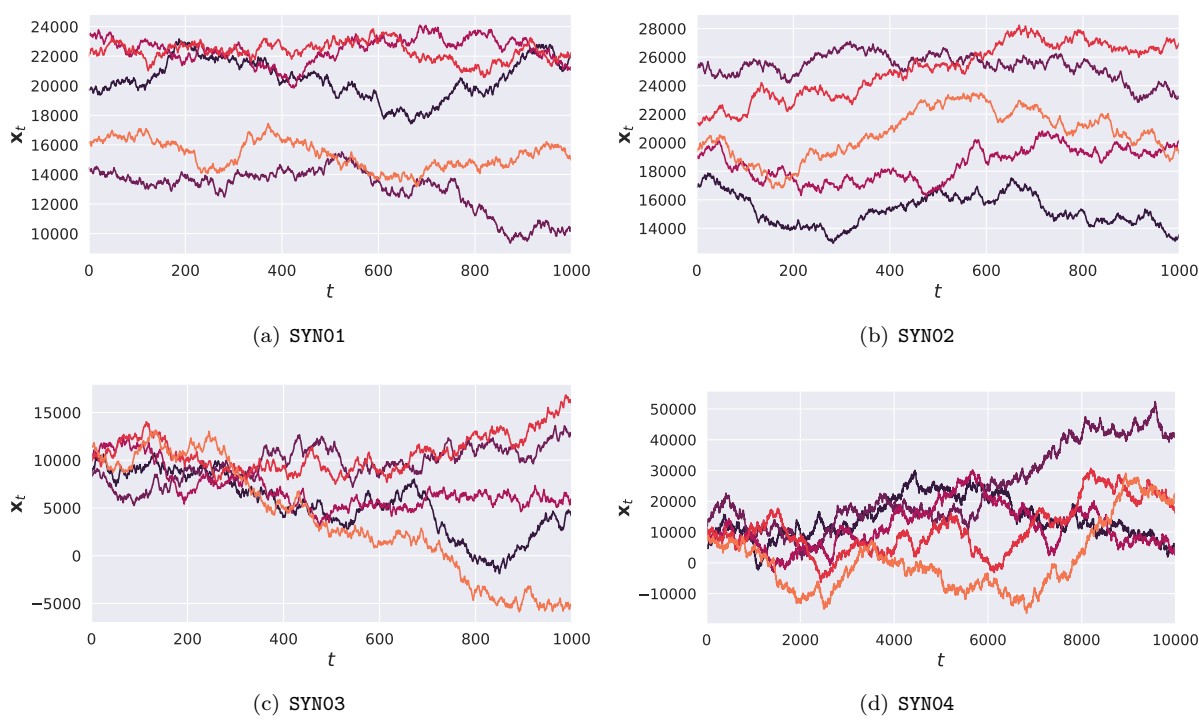

(a) SYN01      (b) SYN02

(c) SYN03      (d) SYN04

Figure 10: Exemplary synthetic dataset samples shown for 5 nodes.

## C.1 Periodicity

The generator parameters for SYN01 and SYN02 are same except for the periodic component added to SYN01 which has a period of $\tau = 100$ timesteps consisting of shocks sampled from $\mathcal{N}(100, 20)$. An algorithm which can exploit the periodic influence in the signal should perform better on SYN01 compared to SYN02. The models which perform worse on periodic dataset are marked red.

Table 6: Impact of data periodicity on RMSE achieved by different models.

| | SYN01 | | | SYN02 | | | % increase |
|---|---|---|---|---|---|---|---|
| | mean | | std. dev. | mean | | std. dev. | $\left(\frac{\text{SYN02}-\text{SYN01}}{\text{SYN01}}\right)$ |
| mspace-S$\mu$ | 299.18 | $\pm$ | 6.55 | 294.99 | $\pm$ | 8.81 | $-0.63$ |
| mspace-S$\mathcal{N}$ | 400.99 | $\pm$ | 3.74 | 395.33 | $\pm$ | 3.24 | $-1.52$ |
| STGODE | 420.86 | $\pm$ | 103.29 | 420.25 | $\pm$ | 52.17 | $-9.87$ |
| GRAM-ODE | 921.94 | $\pm$ | 537.63 | 853.77 | $\pm$ | 340.45 | $-18.18$ |
| LightCTS | 419.43 | $\pm$ | 176.5 | 334.59 | $\pm$ | 79.01 | $-30.6$ |
| Kalman-$x$ | 781.94 | $\pm$ | 32.35 | 776.75 | $\pm$ | 30.38 | $-0.88$ |
| Kalman-$\varepsilon$ | 393.76 | $\pm$ | 4.72 | 390.45 | $\pm$ | 3.54 | $-1.13$ |

## C.2 Training Samples

The generator parameters for SYN03 and SYN04 are same except for the total number of samples being ten times more in SYN04. If a model perform better on SYN04 compared to SYN03, it would indicate that it is training intensive, requiring more samples to infer the trends. On the other hand, if the model performs worse on SYN04, it would indicate that there are scalability issues, or the training caused overfitting. An ideal model is expected to have similar performance on SYN03 and SYN04. The models with ideal behaviour are marked teal, and the models susceptible to overfitting are marked red. Moreover, model(s) that require more training samples are marked violet.

Table 7: Impact of number of training samples on RMSE achieved by different models.

| | SYN03 | | | SYN04 | | | % increase |
|---|---|---|---|---|---|---|---|
| | mean | | std. dev. | mean | | std. dev. | $\left(\frac{\text{SYN04}-\text{SYN03}}{\text{SYN03}}\right)$ |
| mspace-S$\mu$ | 793.41 | $\pm$ | 5.86 | 789.36 | $\pm$ | 3 | $-0.86$ |
| mspace-S$\mathcal{N}$ | 793.93 | $\pm$ | 5.73 | 792.61 | $\pm$ | 2.02 | $-0.63$ |
| STGODE | 830.63 | $\pm$ | 127 | 931.33 | $\pm$ | 191.87 | $+17.29$ |
| GRAM-ODE | 1382.48 | $\pm$ | 80.78 | 1423.93 | $\pm$ | 190.13 | $+10.31$ |
| LightCTS | 769.34 | $\pm$ | 196.6 | 998.01 | $\pm$ | 319.72 | $+36.42$ |
| Kalman-$x$ | 785.7 | $\pm$ | 8.95 | 721.88 | $\pm$ | 1.73 | $-8.94$ |
| Kalman-$\varepsilon$ | 782.6 | $\pm$ | 6.5 | 783.36 | $\pm$ | 1.45 | $-0.54$ |

# D Evaluation

## D.1 Metrics

The root mean squared error (RMSE) of $q$ consecutive predictions for all the nodes is:

$$\text{RMSE}(q) \triangleq \mathbb{E}\left[\sqrt{\frac{1}{ndq}\sum_{v\in\mathcal{V}}\sum_{i\in[q]}\left\|\sum_{j\in[i]}\varepsilon_{t+j}(v) - \hat{\varepsilon}_{t+j}(v)\right\|_2^2}\right]. \tag{22}$$

The mean absolute error (MAE) of $q$ consecutive predictions for all the nodes is:

$$\text{MAE}(q) \triangleq \frac{1}{ndq}\mathbb{E}\left[\sum_{v\in\mathcal{V}}\sum_{i\in[q]}\left\|\sum_{j\in[i]}\varepsilon_{t+j}(v) - \hat{\varepsilon}_{t+j}(v)\right\|_1\right]. \tag{23}$$

## D.2 Datasets

In Table 8, we list the datasets commonly utilised in the literature for single and multi-step node feature forecasting.

**tennis (Béres et al., 2018)** represents a discrete-time dynamic graph showing the hourly changes in the interaction network among Twitter users during the 2017 Roland-Garros (RG17) tennis match. The input features capture the structural attributes of the vertices, with each vertex symbolizing a different user and the edges indicating retweets or mentions within an hour [8].

**wikimath (Rozemberczki et al., 2021a)** tracks daily visits to Wikipedia pages related to popular mathematical topics over a two-year period. Static edges denote hyperlinks between the pages [9].

**pedalme (Rozemberczki et al., 2021a)** reports weekly bicycle package deliveries by Pedal Me in London throughout 2020 and 2021. The nodes are different locations, and the edge weight encodes the physical proximity. The count of weekly bicycle deliveries in a location forms the node feature footnote [10].

**cpox (Rozemberczki et al., 2021b)** tracks the weekly number of chickenpox cases for each county of Hungary between 2005 and 2015. Different counties form the nodes, and are connected if any two counties share a border [10].

**PEMS03/04/07/08 (Rao et al., 2022)** The four datases are collected from four districts in California using the California Transportation Agencies (CalTrans) Performance Measurement System (PeMS) and aggregated into 5-minutes windows[11] . The spatial adjacency matrix for each dataset is constructed using the length of the roads. `PEMS03` is collected from September 2018 to November 2018. `PEMS04` is collected from San Francisco Bay area from July 2016 to August 2016. `PEMS07` is from Los Angeles and Ventura counties between May 2017 and August 2017. `PEMS08` is collected from San Bernardino area between July 2016 to August 2016.

*Variables:* The **flow** represents the number of vehicles that pass through the loop detector per time interval (5 minutes). The **occupancy** variable represents the proportion of time during the time interval that the detector was occupied by a vehicle. It is measured as a percentage. Lastly, the **speed** variable represents the average speed of the vehicles passing through the loop detector during the time interval . It is measured in miles per hour (mph).

**PEMSBAY (Li et al., 2018)** is a traffic dataset collected by CalTrans PeMS. It is represented by a network of 325 traffic sensors in the Bay Area with 6 months of traffic readings ranging from January 2017 to May 2017 in 5 minute intervals[12].

**METRLA (Li et al., 2018)** is a traffic dataset based on Los Angeles Metropolitan traffic conditions. The traffic readings are collected from 207 loop detectors on highways in Los Angeles County over 5 minute intervals between March 2012 to June 2012[13].

### D.3 Baselines

**DCRNN (Li et al., 2018)** The Diffusion Convolutional Recurrent Neural Network (`DCRNN`) models the node features as a diffusion process on a directed graph, capturing spatial dependencies through bidirectional random walks. Additionally, it addresses nonlinear temporal dynamics by employing an encoder-decoder architecture with scheduled sampling.

**TGCN (Zhao et al., 2019)** Temporal Graph Convolutional Network (`TGCN`) combines the graph convolutional network (GCN) with a gated recurrent unit (GRU), where the former learns the spatial patterns, and the latter learns the temporal.

---

[8] https://github.com/ferencberes/online-centrality
[9] `wikimath` dataset from PyTorch Geometric Temporal
[10] https://github.com/benedekrozemberczki/spatiotemporal_datasets
[11] https://github.com/guoshnBJTU/ASTGNN/tree/main/data
[12] `PEMSBAY` dataset from PyTorch Geometric Temporal
[13] `METRLA` dataset from PyTorch Geometric Temporal

Table 8: Real world datasets for single and multi-step forecasting.

| Name | $n$ | $x$ | time-step | $T$ |
|---|---|---|---|---|
| tennis | 1,000 | # tweets | 1 hour | 120 |
| wikimath | 1,068 | # visits | 1 day | 731 |
| pedalme | 15 | # deliveries | 1 week | 35 |
| cpox | 20 | # cases | 1 week | 520 |
| PEMS03 | 358 | flow | 5 min | 26,208 |
| PEMS04 | 307 | flow, occupancy, speed | 5 min | 16,992 |
| PEMS07 | 883 | flow | 5 min | 28,224 |
| PEMS08 | 170 | flow, occupancy, speed | 5 min | 17,856 |
| PEMSBAY | 325 | speed | 5 min | 52,116 |
| METRLA | 207 | speed | 5 min | 34,272 |

**EGCN (Pareja et al., 2020)** EvolveGCN (`EGCN`) adapts a GCN model without using node embeddings. The evolution of the GCN parameters is learnt through an RNN. `EGCN` has two variants: `ECGN-H` which uses a GRU, and `ECGN-O` which uses an LSTM.

**DynGESN (Micheli & Tortorella, 2022)** Dynamic Graph Echo State Networks (`DynGESN`) employ echo state networks (ESNs) a special type of RNN in which the recurrent weights are conditionally initialized, while a memory-less readout layer is trained. The ESN evolves through state transitions wheere the states belong to a compact space. For more details please refer to the original text.

**GWNet (Wu et al., 2019)** GraphWave Net (`GWNet`) consists of an adaptive dependency matrix which is learnt through node embeddings, which is capable of capturing the hidden spatial relations in the data. `GWNet` can handle long sequences owing to its one-dimensional convolutional component whose receptive field grows exponentially with the number of layers.

**STGODE (Fang et al., 2021)** Spatial-temporal Graph Ordinary Differential Equation (`STGODE`) employs tensor-based ordinary differential equations (ODEs) to model the temporal evolution of the node features.

**GRAM-ODE (Liu et al., 2023)** Graph-based Multi-ODE (`GRAM-ODE`) improves upon `STGODE` by connecting multiple ODE-GNN modules to capture different views of the local and global spatiotemporal dynamics.

**FOGS (Rao et al., 2022)** `FOGS` utilises first-order gradients to train a predictive model because the traffic data distribution is irregular.

**LightCTS (Lai et al., 2023)** `LightCTS` stacks temporal and spatial operators in a computationally-efficient manner, and uses lightweight modules L-TCN and GL-Former.

**ARIMA (Box & Pierce, 1970)** `ARIMA` is a multivariate time series forecasting technique that combines autoregressive, integrated, and moving average components. It models the relationship between observations and their lagged values, adjusts for non-stationarity in the data, and accounts for short-term fluctuations.

**Kalman (Welch, 1997)** Since `mspace` is a state-space algorithm, we also use the Kalman filter as a baseline. We introduce two variants of the Kalman filter: `Kalman-`$x$, which considers the node features as observations, and `Kalman-`$\varepsilon$, which operates on the shocks.

## E  Interpretability

The poor performance of `mspace-T`$\mu$ on the datasets `PEMSBAY` and `METRLA` is explained through Fig. 11. We notice that there are many datapoints away from the mean, although the mean trend passed through the dense collection of data points, i.e., the variance in the data is high which leads to higher error values reported in Table 3.

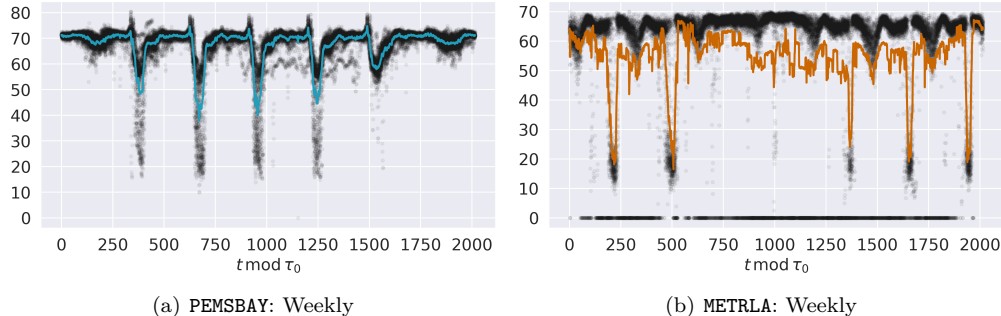

(a) `PEMSBAY`: Weekly                    (b) `METRLA`: Weekly

Figure 11: Periodic trends in the traffic dataset `PEMSBAY` and `METRLA`; the black points represent the datapoints, and the red line is the mean estimate for each state $t \bmod \tau_0$.

## F  Runtime

In Table 9, we present the average execution time of different models. It must be noted that `mspace` was not optimized for GPUs.

Table 9: Average execution time per time-step ($\log_{10}$ scale).

| | TRAIN | | | | TEST | | | |
|---|---|---|---|---|---|---|---|---|
| **CPU** | tennis | wikimath | pedalme | cpox | tennis | wikimath | pedalme | cpox |
| DynGESN | -3.73 | -2.93 | -4.02 | -4.28 | -3.33 | -2.40 | -3.68 | -3.83 |
| ECGN-H | -0.79 | -0.56 | -1.34 | -1.32 | -2.66 | -2.50 | -2.95 | -2.93 |
| ECGN-O | -0.85 | -0.60 | -1.56 | -1.57 | -2.86 | -2.62 | -3.22 | -3.24 |
| TGCN | -0.57 | -0.12 | -1.18 | -1.17 | -2.40 | -2.50 | -2.81 | -2.82 |
| mspace-S$\mu$ | -2.27 | -1.54 | -3.53 | -3.59 | -1.49 | -0.30 | -3.25 | -3.07 |
| mspace-S$\mathcal{N}$ | -2.24 | -1.53 | -3.53 | -3.61 | -1.15 | -0.21 | -2.63 | -2.72 |
| LightCTS | 1.18 | 1.18 | -0.91 | -1.15 | -1.21 | -1.32 | -3.24 | -3.70 |
| STGODE | 0.46 | 0.51 | -0.55 | -0.68 | -2.06 | -2.02 | -2.85 | -3.22 |
| GRAMODE | 1.25 | 1.26 | -0.64 | -0.90 | -0.87 | -0.82 | -2.44 | -2.86 |
| **GPU** | tennis | wikimath | pedalme | cpox | tennis | wikimath | pedalme | cpox |
| DynGESN | -3.99 | -3.66 | -2.61 | -3.65 | -3.52 | -3.50 | -3.63 | -3.65 |
| ECGN-H | -0.79 | -0.78 | -1.00 | -1.00 | -2.36 | -2.34 | -2.60 | -2.59 |
| ECGN-O | -1.22 | -1.21 | -1.28 | -1.30 | -2.91 | -2.90 | -2.92 | -2.92 |
| TGCN | -0.88 | -0.81 | -0.94 | -0.92 | -2.53 | -2.48 | -2.54 | -2.53 |
| LightCTS | 0.11 | 0.13 | -0.95 | -1.41 | -2.34 | -2.40 | -3.32 | -3.98 |
| STGODE | -0.37 | -0.36 | -0.62 | -0.91 | -2.76 | -2.85 | -2.53 | -3.25 |
| GRAMODE | 0.26 | 0.29 | -0.71 | -1.17 | -1.81 | -1.79 | -2.32 | -2.94 |

