# OpenReview forum: "Node Feature Forecasting in Temporal Graphs: an Interpretable Online Algorithm"
_TMLR — Accepted by TMLR_

### Review · Reviewer_yvAf · 2025-02-16

**Summary Of Contributions:**

The paper proposes a graph node value forecasting approach dubbed **mspace**. The approach is modeling the shocks as a Markov process, which is approximated using state functions, which can capture spatial and/or temporal aspects.

One contribution is that due to the model learning Gaussian distributions it can make predictions in both deterministic and probabilistic modes, by either using mean of the distribution or sampling from it - thus being applicable in generative as well as discriminative settings.

Also, method is online and its computation grows linearly in number of nodes and in the number of timesteps. Error in terms of RMSE is also growing linearly in the number of steps.

Interpretability is highlighted as another beneficial aspect of the approach, as predictions can be explained by inputs and model parameters.

Finally, an empirical study was conducted on multiple datasets against multiple competing approaches and the quantitative results are suggesting that proposed approach is typically among the top few performing approaches.

**Audience:**

Yes

**Claims And Evidence:**

Yes

**Requested Changes:**

In the introduction it was claimed that TGNN models are trained to forecast predetermined number of steps, and for any other number of steps they would need to be reinitialized and retrained. However, any forecasting model can be be applied recursively (aka iterative multistep) on arbitrary number of steps - without the necessity of retraining.

**Strengths And Weaknesses:**

Strengths:
- Clearly written paper, well structured and goes into details required for reproducibility
- Lightweight, computationally efficient, yet adaptive to trends and distribution changes
- Provides theoretical results like computational complexity analysis and error bounds

Weaknesses:
- For $\Psi_{S}$ statespace can grow exponentially large the bigger the neighborhood of the node is considered
- Its unclear and not much explored why Kalman Filter applied on shocks is performing so poorly

---

> ### Author Response · Authors · 2025-02-19
>
> We thank the reviewer for their time and effort in reviewing our manuscript.
>
> **Comment on Weaknesses**
> - As stated in lines 137-138, for large neighborhoods, the state space grows prohibitively large. We have acknowledged this limitation in the paper.
> - We thank the reviewer for their comment regarding Kalman filter applied to shocks. We will incorporate this point in the Conclusion as a potential direction for future work.
>
> **Comment on Requested Changes**
> We agree with the reviewer that TGNN models can be fed their output recursively to forecast more steps than they are trained for. We will add this during revision of the manuscript. Thank you!
>
> We hope this addresses reviewer's concerns.

---

### Review · Reviewer_UPbb · 2025-02-17

**Summary Of Contributions:**

The summary of the contributions of the paper is as follows:

The paper introduces mspace, an online algorithm for forecasting node features in discrete-time temporal graphs that captures both spatial cross-correlation and temporal auto-correlation. It supports both probabilistic and deterministic multi-step forecasting and outperforms several baselines, including Temporal Graph Neural Networks (TGNNs) and Kalman filters, especially in scenarios with limited training data. The algorithm’s forecasting error scales linearly with the number of forecast steps, and its computational complexity grows linearly with nodes and time steps, while maintaining constant space complexity. Comparative experiments on ten real-world datasets show mspace’s strong performance, and its interpretability is analyzed.

**Audience:**

Yes

**Broader Impact Concerns:**

There are no "Broader Impact Concerns" for this paper.

**Claims And Evidence:**

No

**Requested Changes:**

Based on the aforementioned strengths and weaknesses, the requested changes are provided as follows:

- Restructuring the entire paper specially the sections such as abstract and conclusion to help improve the language of the paper for the audience to understand the contribution of the paper. The ordering of the sections is very difficult to follow and it heavily impact the clarity of narration.

- The introduction does not motivate and highlight the problem. It is very hard to understand what and why this paper is contributing to.

- There are unsupported statements either needs references or proves to justify (e.g. line 39-48).

- The comparison with the baseline across various q values is insightful.

- The comparison with the baselines to show empirically the interpretability and efficiency improvement of the work is needed.

**Strengths And Weaknesses:**

## Strengths
- the paper proposes an approach for node regression with controlled computational complexity.

- The formulation of the method is simple and based on probabilistic approached.

- The analysis on the interpretability of the method is helpful.


## Weakness

- the claim is not clear. It is not straightforward neither from abstract nor introduction what is the claim of the paper is. It seems overall the paper has a bag of contributions but it is not justified how they uniformly contribute to a primary particular significant contribution that would impact a particular problem. For instance it is not clear if the claim is that the proposed work is to have a node regression model for temporal graphs which has improved time and space complexity or to tackle the limited training data scenario.

- the clarity of the narrative and arguments is not satisfied:
    - The paper structure needs further restructuring. The abstract does not follow a standard composition in which problem is defined. It is extremely difficult to understand the paper. In a standard format, the abstract should decompose into the following structure: Why is it relevant? why is it hard? Indicate the significance of our work? How does it solved? What is the contribution? How do we verify we solved it? Is it empirical? What is the experimental setup? Elaborate on the empirical evidence supporting we solve it; main table, ablation studies, analysis.

    - It is not clear by looking at the abstract and introduction why the problem is and why it is important to solve the problem. It is very weakly motivated on the significance of the research and the necessity of the solution. For instance, the introduction begins with a paragraph that introduce the temporal graphs and then in the second paragraph sudden the contribution is given.

    - It is strange the related work section is between the methodology and results.

    - The related work is very unusually formatted. It has introductory content and notations presented. There are material contamination from the background and problem formulation in the related work.

- the evidence provided in the experimental setup is not convincing that the proposed method is on par with the baselines as claimed in the abstract. For instance in table 3, there case several cases the results of the proposed method falls behind the top ones. So it stronger to say the results are mixed rather than on par. Therefore, the evidence is not convincing enough. In Fig. 5, it is also depicted.

- It is not clear how the interpretability of this method is different from the baselines. The evidence to support it is missing.

- the computation complexity of this paper is not compared with the baselines hence missing evidence.

- The experimental setup seems very simple and the scale of the datasets seems falling behind to accurately support the claims of the paper.

---

> ### Author Response · Authors · 2025-02-19
> **In response to comments on the Abstract**
>
> **[COMMENT] The claim is not clear. It is not straightforward neither from abstract nor introduction what is the claim of the paper is. It seems overall the paper has a bag of contributions but it is not justified how they uniformly contribute to a primary particular significant contribution that would impact a particular problem. For instance it is not clear if the claim is that the proposed work is to have a node regression model for temporal graphs which has improved time and space complexity or to tackle the limited training data scenario.**
>
> ---
>
> We thank the reviewer for their time and effort.
>
> We agree with the reviewer that our work has *multiple contributions*. However, we are surprised to see this being mentioned as a weakness instead of strength.
>
> To clarify, the abstract explicitly highlights the key aspects of our work:
> - we propose an online algorithm mspace for forecasting node features in temporal graphs (line 1)
> - employing mspace is advantageous in scenarios where the training sample availability is limited (line 10)
> - multi-step forecasting error of mspace and show that it scales linearly with the number of forecast steps (line 12)
> - we compare the performance of various mspace variants against ten recent TGNN baselines across ten real-world datasets (line 16)
> - we have investigated the interpretability of different mspace variants by analyzing model parameters alongside dataset characteristics to jointly derive model-centric and data-centric insights (line 17)
>
> As the title suggests, the focus of our work is **interpretability**, which encompasses (1) theoretical error bounds, (2) computational and space complexity, and (3) a data-centric analysis of model performance. A detailed discussion on interpretability is provided in Section 6 (lines 285–346).
>
> Additionally, the contributions are explicitly stated in the Introduction (lines 49–69), reinforcing the coherence of our work. We hope this clarification addresses the reviewer’s concern.

---

> ### Author Response · Authors · 2025-02-19
> **Claims and Evidence**
>
> In this work, we have provided evidence for all the claims. We will demonstrate that in the following table.
>
> | Claim | Evidence |
> |---|---|
> | mspace can sequentially predict the node features for q future timestamps after observing only two past nodes | Algorithm 1 |
> | mspace can produce both probabilistic and deterministic forecasts | lines 143 - 145 |
> | The RMSE for q-step forecast scales linearly in q | App. A  |
> | For asymptotically large number of nodes n and timesteps T, the computational complexity grows linearly both in n and T and the space complexity is constant | App. B |
> | The performance of mspace is compared with 10 TGNN models and 2 classical baselines on 10 datasets | Sec. 5 |
> | We have explained the results of mspace through the data | Sec. 6 |
> | We have provided the code for reproducibility | link in paper |
>
> ---
>
> What we did **NOT** claim in the paper:
> 1. We did not claim that the computation complexity of mspace is better than the baselines. Therefore, there is no need of evidence.
> 2. We did not make any bold claims regarding the performance of mspace on very large datasets, except theoretically for which proofs are already provided as evidence. For the experiments, we have transparently presented the size of the datasets in Table 1 and then compared the performance of mspace with the baselines on them.
>
> ---
>
> *Regarding the interpretability of the TGNN baselines:*
> We have carefully reviewed the TGNN baselines cited in this manuscript and found that they do not emphasize interpretability. However, if the reviewer is aware of evidence to the contrary, we would greatly appreciate any specific references or insights they can share.
>
> ----
>
> *Regarding the phrase at par:* We used at par to describe the performance of mspace wrt the TGNN models overall. However, we agree with the reviewer that this might be confusing and suggest that the performance is the same as TGNN models. Therefore, in the revision, we will replace **at par with SoTA** with **comparable to the TGNN baselines** as suggested by the results in Tables 2 and 3 (summary presented below).
>
>
> | Method | tennis | wikimath | pedalme | cpox | PEMS03 | PEMS04 | PEMS07 | PEMS08 | PEMSBAY | METRLA |
> |---|---|---|---|---|---|---|---|---|---|---|
> | TGNN (best) | 150.41 | 279.87 | 0.91 | 0.83| 24.09 | 30.14 | 33.96 | 23.49 | 3.34 | 6.64 |
> | TGNN (worst) | 206.50 | 1137.68 | 1.58 | 1.07 | 32.94 | 39.70 | 42.78 | 31.05 | 4.89 | 7.81 |
> | mspace (best)| 105.32 | 563.69 | 0.86 | 1.58 | 26.53 | 13.49 | 38.83 | 10.35 | 3.77 | 10.08 |
>
>
> ---
>
> As outlined in this response, all claims in the paper are well-supported by evidence. We kindly request the reviewer to reconsider their evaluation of **Claims and Evidence** and mark it as *Yes*.

---

> > ### Comment · Reviewer_UPbb · 2025-02-25
> > **there are still unaddressed concerns**
> >
> > Thanks for the through and detailed responses. As noted by the evaluation criteria of TMLR, the the claims made in the submission needs to be supported by _accurate_, _convincing_ and _clear_ evidence. As noted before, the paper currently needs significant improvement on two aspects: 1) the narration of the claims are not clear, unified, and untangled across the paper. Additionally, there is a lack of accurate, convincing, and clear evidence for the list of claims.
> >
> > On the former, to emphasize on the unclear twisted narration of the claim of the paper, the response by the authors at some point states that "As the title suggests, the focus of our work is interpretability, ...". However, walking through the abstract, this core contribution is not properly stated and justified not until the last part of the abstract on line 17 where the they begin saying "Lastly, we have investigated the interpretability of different mspace variants ...". This issue is widespread across the different sections of the paper and needs major revision to be addressed. Additionally, there is a lack of motivation on why this is an issue for the research community and what the significance of the research is and how this paper would contribute to community.
> >
> > Additionally, there is a twist in this sentence in the abstract "Importantly, mspace demonstrates consistent performance across datasets with varying training sizes, a notable advantage over TGNN models that require abundant training samples to effectively learn the spatiotemporal trends in the data. Therefore, employing mspace is advantageous in scenarios where the training sample availability is limited.". What is the evidence to support the claim "varying training sizes". This is not precise as the range of benchmark datatsets are limited to datasets with 1K nodes & 100 interactions to 300 nodes and 50K. It is vague what is a notable advantage over TGN models, being consistent across this range of datasets, or having higher performance on very small datasets. As noted before based on Figure 5, mspace-S_mu is not good in terms of performance for larger datasets. Therefore the claim is not valid. mspace-T_mu seems has a better case but it is not clear why there is sudden spike on the one before last and suddenly it drops for the largest one. The conclusion that "Therefore, employing mspace is advantageous in scenarios where the training sample availability is limited." is not supported by a precise accurate and clear evidence. To show that, the results of the following experiments can be insighfull. For the largest dataset, vary the partitioning/split ratio of the training and test set and then show the comparison of the mspace with the baselines. Then the analysis of the results should be more supportive of the claim.
> >
> > Missing evidence to support the claim for interpretability. There is no discussion and comparison on the interpretability of baseline methods. If the claim is the proposed method is more interpretable, there should be evidence showing the predictions of the baseline is not interpretable or not at a comparable level. The tries to support for the interpretability of mspace-S_mu on line 320 using the improvement on 2 the smallest datasets while there is no comment on why the performance degrades on the largesr ones and how that impact the interpretablity.
> >
> > There is a counterintuitve observarion that the proposed method falls behind the TGNN models on small datasets for single-step forecasting (second on two and below thirds on the other). It is insightgul the authors elaborate on given this observation, how the results support the claim "Therefore, employing mspace is advantageous in scenarios where the training sample availability is limited".
> >
> > The following experiments are interesting to add:
> > - The comparison with the baseline across various q values is insightful.
> >
> > The comparison of the time and space complexity of the proposed method with respect to the baseline methods are missing.

---

> > > ### Author Response · Authors · 2025-02-25
> > >
> > > We thank the reviewer for their feedback and appreciate the opportunity to clarify our response.
> > >
> > > As shown in the table in our response titled **Claims and Evidence**, we have provided evidence for all the claims, thereby meeting the evaluation criteria of TMLR. We would be happy to clarify any further objective questions the reviewer may have.
> > >
> > >
> > > **Motivation:** In our revision, we will include a paragraph further clarifying the motivation behind our work. However, we would like to note that this aspect does not fall under the grounds for Claims and Evidence.
> > >
> > >
> > > **Training samples:** We would like to clarify that we have used temporal graphs, where node features change over discrete time points. When referring to training size, we specifically mean the number of timesteps (T) in Table 1. As mentioned in line 253, the train-to-test split is 8:2. Furthermore, in Fig. 5, we have illustrated the number of samples (training size) on the x-axis and the corresponding relative performance on the y-axis. We appreciate the reviewer’s suggestion for an additional experiment. However, we believe that Fig. 5 already provides valid evidence supporting our claim that mspace performs well when training samples are limited.
> > >
> > > Regarding the performance of mspace-S on different datasets, we have provided an explanation in Fig. 8 (page 10). As requested, we will include a diagram in the appendix to further explain why the performance of mspace-T on METRLA worsens.
> > >
> > >
> > > Regarding **Interpretability**, we have already addressed this concern in our response:
> > > > We have carefully reviewed the TGNN baselines cited in this manuscript and found that they do not emphasize interpretability. However, if the reviewer is aware of evidence to the contrary, we would greatly appreciate any specific references or insights they can share.
> > >
> > > Additionally, we kindly ask the reviewer to refer to our previous response, where we clarified what we have **not claimed** in the manuscript. As such, no further evidence is necessary for those points.
> > >
> > > ---
> > >
> > > Lastly, we would like to reiterate that all claims in the paper are well-supported by evidence. We sincerely request the reviewer to reconsider their evaluation of **Claims and Evidence** and mark it as **Yes**.

---

> ### Author Response · Authors · 2025-02-19
> **Clarifying Lines  39-48**
>
> The reviewer commented: **There are unsupported statements either needs references or proves to justify (e.g. line 39-48).**
>
> ---
>
> `39-42:` **If the test data distribution differs from the training data, an offline model cannot adapt. Therefore, when dealing with time-series data, it is crucial to use a lightweight online algorithm that can adapt to changes in data distribution while also performing forecasts.**
>
> In the revision, we will cite: Wang, Liyuan, et al. "A comprehensive survey of continual learning: theory, method and application." IEEE Transactions on Pattern Analysis and Machine Intelligence (2024).
>
> `42-43:` **Moreover, TGNN models are typically trained to forecast a predetermined number of future steps. If we want to increase the number of forecast steps, even by one, the model needs to be reinitialized and retrained.**
>
> This can be verified from the source code of the TGNN models cited in the manuscript. However, as kindly pointed out by reviewer **yvAf**, we can generate output from the TGNN recursively by feeding the output back as input. We will revise the manuscript to reflect this.
>
> `45-46:` **Inspired by the simplicity of Markov models, we define the state of a graph at a given time in an interpretable manner and propose a lightweight model that can be deployed without any training.**
>
> This is explained in Algorithm 1, where the Offline training part is optional. For a detailed discussion on interpretability, please see Sec. 5.
>
> `46-48:` **The algorithm is designed with a mechanism to prioritize recent trends in the data over historical ones, allowing it to adapt to changes in data distribution.**
>
> We kindly ask the the reviewer to refer to Fig. 2 in Sec. 3. Since the queues are of finite size, they will pop out the older values when they receive the new ones, thereby prioritizing recent trends over historical ones.

---

### Review · Reviewer_96mB · 2025-03-01

**Summary Of Contributions:**

The authors propose the mspace algorithm for forecasting multiple time series that are connected by a static unweighted graph. The authors formulate the problem by modeling the shock (difference in time series values over consecutive time steps) using a continuous-state Markov chain. Then, they approximate the Markov chain using a state function $\\Psi$ and a sampling function $\\Omega$ to map to and from a finite set of states. They use fixed length queues to collect most recent shocks following a given state. They demonstrate competitive forecasting error compared to temporal graph neural network (TGNN) models and classical algorithms such as Kalman filters. They also provide some evidence of the interpretability of their approach.

**Audience:**

Yes

**Broader Impact Concerns:**

No statement is present, and I don't believe that one is needed.

**Claims And Evidence:**

No

**Requested Changes:**

Major issues:
- A discussion and comparison against recent research on multivariate time series forecasting with graph structure, such as NAR models. If the authors believe that these models are not relevant, then I would expect a clear discussion of why.
- Line 103 to the end of Section 2 contains no references, so it appears as if this entire portion is the contribution of the authors. If this is the case, then it should be stated. If not, then the authors should indicate which portions are novel and which portions are not.
- Figure 8 and corresponding discussion on interpretability: Why does $tr(\\Sigma(s))$ being left skewed explain the strong performance of mspace? The discussion on interpretability should make it clear what the interpretation is and why.

Minor issues:
- Code repository at https://anonymous.4open.science/r/mspace-TMLR is expired so I could not review it.
- The offline training portion is described as optional in Section 6.4. If this is not done, then how do you compute the MLEs for $\\mu_v(s)$ and $\\Sigma_v(s)$? Aren't the queues then empty for every single state?

**Strengths And Weaknesses:**

Strengths:
- Simple approach for forecasting time series over a graph. The mspace-S version of the model, in particular, uses a very simple state function that is just the sign of the of the shocks of neighboring nodes at potentially multiple hops.
- The mspace algorithm is scalable and adaptive to changes in the data distribution and does not need to be re-trained to forecast at different time steps into the future.

Weaknesses:
- No discussion of vector autoregressive (VAR) models. The authors should be comparing their approach to recent research on network vector autoregressive (NAR) models, beginning with Zhu et al. (2017). These NAR models, and later variants including graph VARMA and graph GARCH models (Hong et al., 2023), are ideally suited to the proposed task.
- Novelty is unclear. Is the approach of mapping the continuous-state Markov chain into finite states using $\\Psi$ and $\\Omega$ itself novel? Or just the choices of the state and sampling functions?
- Discussion of interpretability is unclear. The authors try to explain the failure of their approach using normalized histograms of $tr(\\Sigma(s))$, but there are some gaps in the reasoning.

References:
- Xuening Zhu. Rui Pan. Guodong Li. Yuewen Liu. Hansheng Wang. "Network vector autoregression." Ann. Statist. 45 (3) 1096 - 1123, June 2017. https://doi.org/10.1214/16-AOS1476
- J. Hong, Y. Yan, E. E. Kuruoglu and W. K. Chan, "Multivariate Time Series Forecasting With GARCH Models on Graphs," in IEEE Transactions on Signal and Information Processing over Networks, vol. 9, pp. 557-568, 2023, doi: 10.1109/TSIPN.2023.3304142.

---

> ### Author Response · Authors · 2025-03-02
> **Claims and Evidence**
>
> We thank the reviewer for their time and effort, and valuable feedback.
>
> In this work, we have provided evidence for all the claims, as demonstrated in the following table:
>
> | Claim | Evidence |
> |---|---|
> | mspace can sequentially predict the node features for q future timestamps after observing only two past nodes | Algorithm 1 |
> | mspace can produce both probabilistic and deterministic forecasts | lines 143 - 145 |
> | The RMSE for q-step forecast scales linearly in q | App. A  |
> | For asymptotically large number of nodes n and timesteps T, the computational complexity grows linearly both in n and T and the space complexity is constant | App. B |
> | The performance of mspace is compared with 10 TGNN models and 2 classical baselines on 10 datasets | Sec. 5 |
> | We have explained the results of mspace through the data | Sec. 6 |
> | We have provided the code for reproducibility | link in paper $\star$ |
>
> As outlined in this response, all claims in the paper are well-supported by evidence. We kindly request the reviewer to reconsider their evaluation of **Claims and Evidence** and mark it as *Yes*.
>
> ----
>
> **Link to Code Repository**:
>
> $\star$ The anonymized version recently expired, though it was still accessible at the time of resubmission. For the final version, we will include a direct GitHub link that does not expire. In the meantime, the code can be accessed at https://anonymous.4open.science/r/mspace-TMLR2/. Additionally, we would like to highlight that in our original submission, reviewer `fNZ3` **verified the reproducibility of our results**, and the code has remained unchanged since then.

---

> ### Author Response · Authors · 2025-03-02
> **Interpretability**
>
> [`Fig. 8`] In Fig. 8, we do not necessarily explain the *failure* of mspace, rather, we explain its relative performance on different datasets. When estimating samples from a normal distribution, the error is directly related to the variance of the distribution which is represented through the trace of the covariance matrix. We show that for datasets, the histogram of whose variance (trace of covariance matrix) is closer to zero, the performance of mspace is relatively better. On the contrary, if the histogram has a thick tail (see PEMSBAY and METRLA), we can expect the performance to be worse. We will clarify this explanation further in Sec. 6.1 when we post the **revision**.
>
> [`Empty queues`] The reviewer highlighted that since training is optional, how would the MLEs be computed for states that were never observed before. In Algorithm 1, in line 19, we show that s* is found to be the one closest to Psi(previous shock). Therefore, if a new state is encountered, the algorithm will use data associated with a state closest to it in memory. As more data is acquired, we will find a  more closely matching state (and perhaps the same state). In the case of no-match, which will only happen for the first step, a zero-value is returned. We thank the reviewer for pointing this out, and we will mention this in the **revision**.

---

> ### Author Response · Authors · 2025-03-02
> **Misc. Comments related to Novelty**
>
> [`Novelty`] **Novelty is unclear. Is the approach of mapping the continuous-state Markov chain into finite states using Ψ and Ω  itself novel? Or just the choices of the state and sampling functions?**
>
> We only claim novelty for defining the state and sampling functions as we did in the paper.
>
> ---
>
> [`References`] **Line 103 to the end of Section 2 contains no references, so it appears as if this entire portion is the contribution of the authors. If this is the case, then it should be stated. If not, then the authors should indicate which portions are novel and which portions are not.**
>
> Since Sec. 2 discusses the methodology and the rationale behind the design of mspace, we consider this text to be our own. If the reviewer has any suggestions regarding citations within Sec. 2, we would greatly appreciate their input.

---

> > ### Comment · Reviewer_96mB · 2025-04-17
> >
> > Since the claimed novelty is only the specific form of the state and sampling functions, it would be helpful for the authors to provide examples of different types of state and sampling functions used in prior work. This could go somewhere between line 123 where these functions are first introduced and line 137 where you start describing the proposed model.

---

> ### Author Response · Authors · 2025-03-03
> **Comparison with Vector Autoregressive Models**
>
> We sincerely thank the reviewer for bringing NAR (Zhu et al., 2017) and Graph-GARCH (Hong et al., 2023) to our attention. These models share similarities with graph convolution approaches, where a node's value is influenced by a weighted sum of its neighbours' values.
>
> We are happy to include a discussion of VAR methods in the related works section in our revision. However, given time constraints, we may not be able to present experimental results on VAR within the main paper. Our current comparisons already include several recent and relevant baselines, such as GRAMODE (2023) and LightCTS (2023). Additionally, as this is a resubmission, none of the other reviewers (in total 6) previously raised concerns about additional baselines.
>
> We believe that including VAR-based results would not alter the key insights of our work, as our comparisons already cover a diverse set of baselines, including classical methods like ARIMA and Kalman filters. Nonetheless, we acknowledge the reviewer's point and can also add a discussion of VAR methods in the limitations section to acknowledge their potential relevance.
>
> Most TGNN baselines cited in this work do not include comparisons with VAR based methods (except DCRNN). For reference, we provide the following results (MAE) from the DCRNN paper:
>
> | Dataset | VAR | DCRNN | Best | mspace-Smu | mspace-Tmu |
> |---|---|---|---|---|---|
> | PEMSBAY |6.52 | 3.60 | 3.42 | 6.56 | 6.77 |
> | METRLA |2.93 | 2.07  | 1.67 | 2.47 | 2.19 |
>
> We hope this will address reviewer's concern, and we thank them for their productive comments.

---

### Comment · Reviewer_M31y · 2025-02-20

In this resubmit version, the authors have addressed several of my previous concerns, particularly in clarifying certain theoretical justifications and refining explanations for key concepts. However, some issues remain, and further improvements would strengthen the clarity and rigor of the paper. Below are my additional comments:

- 1. Justification of Assumption 2.1

The claim that "a continuous-state Markov chain is a weak assumption due to infinite states" is misleading. In high-dimensional spaces (e.g., $\mathbb{R}^{nd}$), the Markov property itself imposes a strong structural constraint, limiting the model's applicability to scenarios with long-range dependencies.

- 2. Undefined Notations and Abrupt Technical Details

When introducing linear dynamical systems and autoregressive models, should first present the high-level motivation behind the model and some key components still lack explanations:
**Lines 103–105:** Sets $\mathcal{C}$ and $\mathcal{S}$, and mapping $\Psi, \Omega$ are directly introduced without definition and explanation.

- 3. Ambiguous Terminology

**Line 33:** "While …, TGNN models could be more interpretable, as …"  What does more interpretable means?
**Line 96:** "Arbitrary set of nodes" conflicts with $\epsilon_t \in \mathbb{R}^{nd}$, which implicitly assumes all nodes.

- 4. State Functions

Should better add a few sentences to briefly explain the intuition of $\Psi_S, \Psi_T$ in Sec.2 and can later be detailedly discussed in Sec.6. Also, it is better to explicitly discuss the limitation of $\Psi_S, \Psi_T$ in Sec.7 as well.

- 5. Invalid Code Link - invalid URL for code base.

---

> ### Author Response · Authors · 2025-02-20
>
> We sincerely thank the reviewer for taking the time to assess our resubmission and for providing thoughtful feedback to further improve our work.
>
> 1. We agree with the reviewer’s point about the Markov assumption limiting the algorithm’s ability to capture long-range dependencies. We will acknowledge this in the revision and explicitly highlight it as a limitation in the Conclusion.
>
>
> 2. We also appreciate the reviewer’s suggestion regarding notation.
> - In the revision, we will mention that $\mathcal{C} \subseteq \mathbb{R}^m$ for some $m \in \mathbb{N}$.
> - We will further describe $\mathcal{S}$ as a finite discrete set in the revision.
> - We have defined $\Psi$ an $\Omega$ as mappings. Equations (1) and (2) can be understood without describing how these mappings are defined. We would like to highlight that in line 114 we have assigned names to these mappings calling them state and sampling functions. Moreover, in lines 139-145 we have defined different variants of these state and sampling functions.
>
> 3. We thank the reviewer for their comment.
> - Line 33: We intended to convey that TGNNs could be more interpretable in terms of explaining their performance through the data – i.e., understanding why a model performs the way it does on a given dataset. We will make this clearer in the revision.
> - Line 96: We confirm that the dimension of $\varepsilon_t$ is not fixed. We will review all instances where it is defined as $\mathbb{R}^{nd}$ and update it to $\mathbb{R}^{md}$ where $m \in \mathbb{N}$.
>
> 4. Following the reviewer’s suggestion, we will add a high-level summary of the two state functions in Sec. 2 and explicitly mention the limitations of our approach in Sec. 7.
>
> 5. Regarding the code link, the anonymized version recently expired, though it was still accessible at the time of resubmission. For the final version, we will include a direct GitHub link that does not expire. In the meantime, the code can be accessed at `https://anonymous.4open.science/r/mspace-TMLR2/`. Additionally, we would like to highlight that in our original submission, **reviewer fNZ3 verified the reproducibility of our results**, and the code has remained unchanged since then.
>
>
> We hope these clarifications address the reviewer’s concerns and that they will feel confident recommending our manuscript for acceptance. We will submit the revised version once all reviews are in.
>
>
> Once again, we deeply appreciate the reviewer’s time and constructive feedback.

---

### Decision · Action_Editor_Jmye · 2025-04-27

**Recommendation:** Accept with minor revision

**Comment:**

This version of the manuscript is significantly improved compared to the previous submission. Nevertheless, some concerns regarding the organization of the results and exposition, and the contributions themselves remain. I am recommending the paper to be accepted in order to reduce processing delays, but strongly advise the authors to take into account the final rounds of comments of the reviewers in order to further strengthen the exposition.

**Audience:**

I believe that the results presented in this work can be of value to a broad TMLR readership.

**Claims And Evidence:**

The revised manuscript has made a good case for acceptance, and to the best of the editor's knowledge the claims made are accurate and sufficiently clear.

---

> ### Author Response · Authors · 2025-04-30
>
> We thank the Action Editor for accepting the manuscript, and all the reviewers for their time and effort. We will further revise the manuscript to incorporate any leftover comments and submit the de-anonymized version by the next week.